# Curvature-based Robustness Certificates against Adversarial Examples

## Abstract

A robustness certificate against adversarial examples is the minimum distance of a given input to the decision boundary of the classifier (or its lower bound). For *any* perturbation of the input with a magnitude smaller than the certificate value, the classification output will provably remain unchanged. Computing exact robustness certificates for deep classifiers is difficult in general since it requires solving a non-convex optimization. In this paper, we provide computationally-efficient robustness certificates for deep classifiers with differentiable activation functions in two steps. First, we show that if the eigenvalues of the Hessian of the network (curvatures of the network) are bounded, we can compute a robustness certificate in the $l_2$ norm efficiently using convex optimization. Second, we derive a computationally-efficient differentiable upper bound on the curvature of a deep network. We also use the curvature bound as a regularization term during the training of the network to boost its certified robustness against adversarial examples. Putting these results together leads to our proposed **C**urvature-based **R**obustness **C**ertificate (CRC) and **C**urvature-based **R**obust **T**raining (CRT). Our numerical results show that CRC outperforms CROWN's certificate when trained with our regularizer while CRT leads to higher certified accuracy compared to standard adversarial training.

## 1 Introduction

Modern neural networks achieve high accuracy on tasks such as image classification and speech recognition, but are known to be brittle to small, adversarially chosen perturbations of their inputs (Szegedy et al., 2014). A classifier which correctly classifies an image $x$, can be fooled by an adversary to misclassify an *adversarial example* $\mathbf{x} + \delta$, such that $\mathbf{x} + \delta$ is indistinguishable from $\mathbf{x}$ to a human. Adversarial examples can also fool systems when they are printed out on a paper and photographed with a smart phone (Kurakin et al., 2016a). Even in a black box threat model, where the adversary has no access to the model parameters, attackers could target autonomous vehicles by using stickers or paint to create an adversarial stop sign that the vehicle would interpret as a yield or another sign (Papernot et al., 2016). This trend is worrisome and suggests that these vulnerabilities need to be appropriately addressed before neural networks can be deployed in security critical applications.

In the last couple of years, several *empirical defenses* have been proposed for training classifiers to be robust against adversarial perturbations (Madry et al., 2018; Samangouei et al., 2018; Zhang et al., 2019; Papernot et al., 2016; Kurakin et al., 2016b; Miyato et al., 2017; Zheng et al., 2016) Although these defenses robustify classifiers to particular types of attacks, they can be still vulnerable against stronger attacks (Athalye et al., 2018; Carlini & Wagner, 2017; Uesato et al., 2018; Athalye & Carlini, 2018). For example, (Athalye et al., 2018) showed most of the empirical defenses proposed in ICLR 2018 can be broken by developing tailored attacks for each of them.

To end the cycle between defenses and attacks, a line of work on *certified defenses* has gained attention where the goal is to train classifiers whose predictions are *provably* robust within some given region (Huang et al., 2016; Katz et al., 2017; Ehlers, 2017; Carlini et al., 2017; Cheng et al., 2017; Lomuscio & Maganti, 2017; Dutta et al., 2018; Fischetti & Jo, 2018; Bunel et al., 2017; Wang et al., 2018a; Wong & Kolter, 2017; Wang et al., 2018b; Wong et al., 2018; Raghunathan et al., 2018b;a; Dvijotham et al., 2018a;b; Croce et al., 2018; Singh et al., 2018; Gowal et al., 2018; Gehr et al., 2018; Mirman et al., 2018; Zhang et al., 2018b; Weng et al., 2018). These methods, however do not scale to large and practical networks used in solving modern machine learning problems. Another line of

defense work focuses on *randomized smoothing* where the prediction is robust within some region around the input with a user-chosen probability (Liu et al., 2017; Cao & Gong, 2017; Lécuyer et al., 2018; Li et al., 2018; Cohen et al., 2019; Salman et al., 2019). Although these methods can scale to large networks, certifying robustness with probability close to 1 often requires generating a large number of noisy samples around the input which leads to high test-time computational complexity.

If the classifier $f(.)$ was linear, the distance of an input point $\mathbf{x}$ to its decision boundary (i.e. the robustness certificate) can be computed efficiently using a convex optimization. For example, the $l_2$ robustness certificate in that case would be equal to $|f(\mathbf{x})|/\|\nabla_\mathbf{x} f(\mathbf{x})\|$. However, modern classifiers based on neural networks are not linear and and can have non-zero curvatures in different parts of the input domain. The deviation of the classifier from the linear model makes the robustness certification problem to be a non-convex optimization which is often difficult to solve exactly. However, *if* we could compute global bounds on the maximum curvature values of the classification network, one may be able to compute computationally-efficient lower bounds on the robustness certificate even for non-linear deep classifiers. This is the key intuition of our results in this paper.

In this work, we derive a global bound on the Lipschitz constant of the gradient of deep neural networks with differentiable activation functions (such as sigmoid, tanh, softplus, etc.). This provides an upper bound on the magnitude of the eigenvalues of the Hessian or the curvature values of the classification network. Using this global curvature bound and for the $l_2$ metric, we tackle both the certification and attack problems. In *the certification problem* and for a given pre-trained classifier, we provide a computationally-efficient lower bound on the distance of a point to the classification decision boundary. In the related *attack problem*, for a given input and a region around it, our goal is to find a perturbed input (an adversarial example) that maximizes the loss inside the given region. The outcome of the attack problem is then used in the adversarial training procedure (Madry et al., 2018) to further robustify the network. Furthermore, our global curvature bound is differentiable and we show that adding it to the loss function as a regularizer boosts *certified robustness* measures.

We note that other recent works (e.g. Moosavi Dezfooli et al. (2019); Qin et al. (2019)) empirically show that using an *estimate* of curvature at inputs as a regularizer leads to *empirical* robustness on par with the adversarial training. In this work, however, we use a provable global upper bound on the curvature (and not an estimate) as a regularizer and show that it results in high *certified* robustness. Moreover, previous works have tried to certify robustness by bounding the Lipschitz constant of the neural network (Szegedy et al., 2014; Peck et al., 2017; Zhang et al., 2018c; Anil et al., 2018; Hein & Andriushchenko, 2017). Our approach, however, is based on bounding the Lipschitz constant of the gradient of deep neural networks. We discuss existing works in more details in Appendix A.

Below, we state the key theoretical results of this paper informally while detailed statements of these results are presented in Section 4.

**Theorem (informal) 1.** *Let $\mathbf{z}_i^{(L)}$ denotes the $i^{th}$ logit of an $L$ layer fully-connected neural network with differentiable activation functions. Then, the curvature of the neural network function is globally bounded as follows:*

$$m\mathbf{I} \preccurlyeq \nabla_\mathbf{x}^2 \mathbf{z}_i^{(L)} \preccurlyeq M\mathbf{I}, \qquad \forall \mathbf{x} \in \mathbb{R}^D$$

*where $m$ and $M$ can be computed efficiently using parameters of the network.*

This result along with the min-max theorem leads to the following curvature robustness certificate:

**Theorem (informal) 2.** *Consider a network whose curvature values are bounded. For a given input $\mathbf{x}^{(0)}$ with the true label $y$ and the attack target $t$ ($t \neq y$), let $p_{cert}^*$ denote the exact robustness certificate, i.e. the distance of $\mathbf{x}^{(0)}$ to the decision boundary. We can efficiently compute $d_{cert}^*$ such that $d_{cert}^* \geq p_{cert}^*$. Moreover, if the solution $\mathbf{x}^{(cert)}$ for $d_{cert}^*$ satisfies $\mathbf{z}_y^{(L)} = \mathbf{z}_t^{(L)}$, then $d_{cert}^* = p_{cert}^*$.*

We have similar results for the attack problem. For simplicity, we summarize definitions of $p_{cert}^*, d_{cert}^*, p_{attack}^*, d_{attack}^*$ in Table 1.

In summary, in this paper, we make the following contributions:

- We provide global bounds on the eigenvalues of the Hessian of a deep neural network with differentiable activation functions (Theorem 3 and Theorem 4). In addition to the adversarial robustness problem, these bounds may be of an independent interest for readers.

| | Certificate problem $_{(-) = cert}$ | Attack problem $_{(-) = attack}$ |
|---|---|---|
| primal problem, $p^*_{(-)}$ | $\min_{f(\mathbf{x})=0} 1/2\|\mathbf{x} - \mathbf{x}^{(0)}\|^2$ | $\min_{\|\mathbf{x}-\mathbf{x}^{(0)}\|\le\rho} f(\mathbf{x})$ |
| dual function, $d_{(-)}(\eta)$ | $\min_{\mathbf{x}} 1/2\|\mathbf{x} - \mathbf{x}^{(0)}\|^2 + \eta f(\mathbf{x})$ | $\min_{\mathbf{x}} f(\mathbf{x}) + \eta/2(\|\mathbf{x} - \mathbf{x}^{(0)}\|^2 - \rho^2)$ |
| When is dual solvable? | $-1/M \le \eta \le -1/m$ | $-m \le \eta$ |
| dual problem, $d^*_{(-)}$ | $\max_{-1/M\le\eta\le-1/m} d_{cert}(\eta)$ | $\max_{-m\le\eta} d_{attack}(\eta)$ |
| When primal = dual? | $f(\mathbf{x}^{(cert)}) = 0$ | $\|\mathbf{x}^{(attack)} - \mathbf{x}^{(0)}\| = \rho$ |

Table 1: A summary of various primal and dual concepts used in the paper. $f$ denotes the function of the decision boundary, i.e. $\mathbf{z}_y^{(L)} - \mathbf{z}_t^{(L)}$ where $y$ is the true label and $t$ is the attack target. $m$ and $M$ are lower and upper bounds on the smallest and largest eigenvalues of the Hessian of $f$, respectively.

| Method | Non-trivial bound | Multi-layer | Activation functions | Norm |
|---|---|---|---|---|
| Szegedy et al. (2014) | ✗ | ✓ | All | $l_2$ |
| Katz et al. (2017) | ✓ | ✓ | ReLU | $l_\infty$ |
| Hein & Andriushchenko (2017) | ✓ | ✗ | Differentiable | $l_2$ |
| Raghunathan et al. (2018a) | ✓ | ✗ | ReLU | $l_\infty$ |
| Wong & Kolter (2017) | ✓ | ✓ | ReLU | $l_\infty$ |
| Weng et al. (2018) | ✓ | ✓ | ReLU | $l_1, l_2, l_\infty$ |
| Zhang et al. (2018b) | ✓ | ✓ | All | $l_1, l_2, l_\infty$ |
| Cohen et al. (2019) | ✓ | ✓ | All | $l_2$ |
| Ours | ✓ | ✓ | Differentiable | $l_2$ |

Table 2: Comparison of methods for providing provable robustness certification. Note that Cohen et al. (2019) is a probabilistic certificate.

- Using the global curvature bounds, we develop computationally efficient methods for both the robustness certification as well as the adversarial attack problems (Theorems 1 and 2).
- We show that using our proposed curvature bounds as a regularizer during training leads to improved certified accuracy on 2,3 and 4 layer networks (on the MNIST dataset) compared to standard adversarial training with PGD (Madry et al., 2018) as well as TRADES (Zhang et al., 2019). Moreover, our robustness certificate (CRC) outperforms CROWN's certificate (Zhang et al., 2018b) significantly while taking less time to compute.

## 2 NOTATION AND PROBLEM SETUP

Consider a fully connected neural network with $L$ layers and $N_I$ neurons in the $I^{th}$ layer ($L \ge 2$ and $I \in [L]$) for a multi-label classification problem with $C$ classes ($N_L = C$). The corresponding function of the neural network is $\mathbf{z}^{(L)} : \mathbf{R}^D \to \mathbf{R}^C$ where $D$ is the dimension of the input. For an input $\mathbf{x}$, we use $\mathbf{z}^{(I)}(\mathbf{x}) \in \mathbf{R}^{N_I}$ and $\mathbf{a}^{(I)}(\mathbf{x}) \in \mathbf{R}^{N_I}$ to denote the input (*before* applying the activation function) and output (*after* applying the activation function) of neurons in the $I^{th}$ hidden layer of the network, respectively. To simplify notation and when no confusion arises, we make the dependency of $\mathbf{z}^{(I)}$ and $\mathbf{a}^{(I)}$ to $\mathbf{x}$ implicit. We define $\mathbf{a}^{(0)}(\mathbf{x}) = \mathbf{x}$ and $N_0 = D$.

With a fully connected architecture, each $\mathbf{z}^{(I)}$ and $\mathbf{a}^{(I)}$ is computed using a transformation matrix $\mathbf{W}^{(I)} \in R^{N_I \times N_{I-1}}$, the bias vector $\mathbf{b}^{(I)} \in R^{N_I}$ and an activation function $\sigma(.)$ as follows:

$$\mathbf{z}^{(I)}(\mathbf{x}) = \mathbf{W}^{(I)}\mathbf{a}^{(I-1)}(\mathbf{x}) + \mathbf{b}^{(I)}, \qquad \mathbf{a}^{(I)}(\mathbf{x}) = \sigma\left(\mathbf{z}^{(I)}(\mathbf{x})\right).$$

We use $(\mathbf{z}_i^{(L)} - \mathbf{z}_j^{(L)})(\mathbf{x})$ as a shorthand for $\mathbf{z}_i^{(L)}(\mathbf{x}) - \mathbf{z}_j^{(L)}(\mathbf{x})$.

We use $[p]$ to denote the set $\{1, \ldots, p\}$ and $[p, q]$, $p \leq q$ to denote the set $\{p, p+1, \ldots, q\}$. We use small letters $i, j, k$ etc to denote the index over a vector or rows of a matrix and capital letters $I, J$ to denote the index over layers of network. The element in the $i^{th}$ position of a vector $\mathbf{v}$ is given by $\mathbf{v}_i$, the vector in the $i^{th}$ row of a matrix $\mathbf{A}$ is $\mathbf{A}_i$ while the element in the $i^{th}$ row and $j^{th}$ column of $\mathbf{A}$ is $\mathbf{A}_{i,j}$. We use $\|\mathbf{v}\|$ and $\|\mathbf{A}\|$ to denote the 2-norm and the operator 2-norm of the vector $\mathbf{v}$ and the matrix $\mathbf{A}$, respectively. We use $|\mathbf{v}|$ and $|\mathbf{A}|$ to denote the vector and matrix constructed by taking the elementwise absolute values. We use $\lambda_{max}(\mathbf{A})$ and $\lambda_{min}(\mathbf{A})$ to denote the largest and smallest eigenvalues of a symmetric matrix $\mathbf{A}$. We use $diag(\mathbf{v})$ to denote the diagonal matrix constructed by placing each element of $\mathbf{v}$ along the diagonal. We use $\odot$ to denote the Hadamard Product, $\mathbf{I}$ to denote the identity matrix. We use $\preccurlyeq$ and $\succcurlyeq$ to denote Linear Matrix Inequalities (LMIs) such that given two symmetric matrices $\mathbf{A}$ and $\mathbf{B}$ where $\mathbf{A} \succcurlyeq \mathbf{B}$ means $\mathbf{A} - \mathbf{B}$ Positive Semi-Definite (PSD).

## 3  USING DUALITY TO SOLVE THE ATTACK AND CERTIFICATE PROBLEMS

Consider an input $\mathbf{x}^{(0)}$ with true label $y$ and the attack target $t$. In the certificate problem, our goal is to find a lower bound of the minimum $l_2$ distance between $\mathbf{x}^{(0)}$ and the decision boundary, $\mathbf{z}_y^{(L)} = \mathbf{z}_t^{(L)}$. The problem for solving the exact distance (*primal*) can be written as:

$$p_{cert}^* = \min_{\mathbf{z}_y^{(L)}(\mathbf{x}) = \mathbf{z}_t^{(L)}(\mathbf{x})} \left[ \frac{1}{2} \left\| \mathbf{x} - \mathbf{x}^{(0)} \right\|^2 \right] = \min_{\mathbf{x}} \max_{\eta} \left[ \frac{1}{2} \left\| \mathbf{x} - \mathbf{x}^{(0)} \right\|^2 + \eta \left( \mathbf{z}_y^{(L)} - \mathbf{z}_t^{(L)} \right)(\mathbf{x}) \right]. \quad (1)$$

However, solving the above problem can be hard in general. Using the minimax theorem (primal $\geq$ dual), we can write the *dual* of the above problem as follows:

$$p_{cert}^* \geq \max_{\eta} \ d_{cert}(\eta), \qquad d_{cert}(\eta) = \min_{\mathbf{x}} \left[ \frac{1}{2} \left\| \mathbf{x} - \mathbf{x}^{(0)} \right\|^2 + \eta \left( \mathbf{z}_y^{(L)} - \mathbf{z}_t^{(L)} \right)(\mathbf{x}) \right]. \quad (2)$$

From the theory of duality, we know that $d_{cert}(\eta)$ for each value of $\eta$ gives a lower bound on the exact certification value (the primal solution) $p_{cert}^*$. However, since $\mathbf{z}_y^{(L)} - \mathbf{z}_t^{(L)}$ is non-convex, solving $d_{cert}(\eta)$ for every $\eta$ can be difficult. In the next section, we will prove that the curvature of the function $\mathbf{z}_y^{(L)} - \mathbf{z}_t^{(L)}$ is bounded globally:

$$m\mathbf{I} \preccurlyeq \nabla_{\mathbf{x}}^2 \left( \mathbf{z}_y^{(L)} - \mathbf{z}_t^{(L)} \right) \preccurlyeq M\mathbf{I} \qquad \forall \mathbf{x} \in \mathbb{R}^D, \ m < 0, \ M > 0 \quad (3)$$

In this case, we have the following theorem:

**Theorem 1.** $d_{cert}(\eta)$ *is a convex optimization problem for* $-1/M \leq \eta \leq -1/m$. *Moreover, If* $\mathbf{x}^{(cert)}$ *is the solution to* $d_{cert}^*$ *such that* $\mathbf{z}_y^{(L)}(\mathbf{x}^{(cert)}) = \mathbf{z}_t^{(L)}(\mathbf{x}^{(cert)})$, *then* $p_{cert}^* = d_{cert}^*$.

Below, we briefly outline the proof while the full proof is presented in Appendix D.1. The Hessian of the *objective function* of the dual $d_{cert}(\eta)$, i.e the function inside the $\min_{\mathbf{x}}$ is given by:

$$\nabla_{\mathbf{x}}^2 \left[ \frac{1}{2} \left\| \mathbf{x} - \mathbf{x}^{(0)} \right\|^2 + \eta \left( \mathbf{z}_y^{(L)} - \mathbf{z}_t^{(L)} \right)(\mathbf{x}) \right] = \mathbf{I} + \eta \nabla_{\mathbf{x}}^2 \left( \mathbf{z}_y^{(L)} - \mathbf{z}_t^{(L)} \right)$$

From equation (3), we know that the eigenvalues of $\mathbf{I} + \eta \nabla_{\mathbf{x}}^2 (\mathbf{z}_y^{(L)} - \mathbf{z}_t^{(L)})$ are bounded between $(1 + \eta m, 1 + \eta M)$ if $\eta \geq 0$, and in $(1 + \eta M, 1 + \eta m)$ if $\eta \leq 0$. In both cases, we can see that for $-1/M \leq \eta \leq -1/m$, all eigenvalues will be non-negative, making the objective function convex. When $\mathbf{x}^{(cert)}$ satisfies $\mathbf{z}_y^{(L)} = \mathbf{z}_t^{(L)}$, $d_{cert}^* = 1/2\|\mathbf{x}^{(cert)} - \mathbf{x}^{(0)}\|^2$, using the duality theorem and definition of $p_{cert}^*$, we get $p_{cert}^* = d_{cert}^*$.

Next, we consider the attack problem. The goal here is to find an adversarial example inside an $l_2$ ball of radius $\rho$ such that $\mathbf{z}_y^{(L)} - \mathbf{z}_t^{(L)}$ is minimized. Using similar arguments, we can get the following theorem for the attack problem ($p_{attack}^*$, $d_{attack}^*$ and $d_{attack}$ are defined in Table 1):

**Theorem 2.** $d_{attack}(\eta)$ *is a convex optimization problem for* $-m \leq \eta$. *Moreover, if* $\mathbf{x}^{(attack)}$ *is the solution to* $d_{attack}^*$ *such that* $\left\| \mathbf{x}^{(attack)} - \mathbf{x}^{(0)} \right\| = \rho$, $p_{attack}^* = d_{attack}^*$.

The proof is presented in Appendix D.2. Both Theorems 1, 2 hold for any non-convex function with continuous gradients. They can also be of interest in problems such as optimization of neural nets.

Using Theorems 1 and 2, we have the following definitions for certification and attack optimizations:

**Definition 1.** *(**Curvature-based Certificate Optimization**) Given an input $\mathbf{x}^{(0)}$ with true label $y$, the false target $t$, we define $(\eta^{(cert)}, \mathbf{x}^{(cert)})$ as the solution of the following max-min optimization:*

$$\max_{-1/M \leq \eta \leq -1/m} \min_{\mathbf{x}} \left[ \frac{1}{2} \left\| \mathbf{x} - \mathbf{x}^{(0)} \right\|^2 + \eta \left( \mathbf{z}_y^{(L)} - \mathbf{z}_t^{(L)} \right)(\mathbf{x}) \right]$$

*We refer to $\left\| \mathbf{x}^{(cert)} - \mathbf{x}^{(0)} \right\|$ as the **C**urvature-based **R**obustness **C**ertificate (CRC).*

**Definition 2.** *(**Curvature-based Attack Optimization**) Given input $\mathbf{x}^{(0)}$ with label $y$, false target $t$, and the $l_2$ ball radius $\rho$, we define $(\eta^{(attack)}, \mathbf{x}^{(attack)})$ as the solution of the following optimization:*

$$\max_{\eta \geq -m} \min_{\mathbf{x}} \left[ \frac{\eta}{2} \left( \left\| \mathbf{x} - \mathbf{x}^{(0)} \right\|^2 - \rho^2 \right) + \left( \mathbf{z}_y^{(L)} - \mathbf{z}_t^{(L)} \right)(\mathbf{x}) \right].$$

*When $\mathbf{x}^{(attack)}$ is used for training in an adversarial training framework, we call the method the **C**urvature-based **R**obust **T**raining (CRT).*

Since both curvature-based certificate and attack optimizations are convex optimization problems, any convex optimization solver can be used to solve them. In our implementation, we use majorization-minimization to solve the dual function for a given $\eta$ and bisection method to maximize over $\eta$. More details are given in Appendix C.4 and C.5.

## 4 Curvature Bounds for deep networks

In this section, we provide a computationally efficient approach to compute the curvature bounds for neural networks with differentiable activation functions. To the best of our knowledge, there is no prior work on finding provable bounds on the curvature values of deep neural networks. Our results rely on a closed form expression for the Hessian of the $i^{th}$ logit as a sum of matrix products (Section 4.1). After establishing this result, we first derive curvature bounds for a two-layer network in Section 4.2 and then extend the bounds to deeper networks in Section 4.3.

### 4.1 Closed form expression for the Hessian

Using the chain rule of second derivatives, we can derive $\nabla_{\mathbf{x}}^2 \mathbf{z}_i^{(L)}$ as a sum of matrix products:

**Lemma 1.** *Given an $L$ layer neural network, the Hessian of the $i^{th}$ hidden unit with respect to the input $\mathbf{x}$, i.e $\nabla_{\mathbf{x}}^2 \mathbf{z}_i^{(L)}$ is given by the following formula:*

$$\nabla_{\mathbf{x}}^2 \mathbf{z}_i^{(L)} = \sum_{I=1}^{L-1} \left( \mathbf{B}^{(I)} \right)^T diag \left( \mathbf{F}_i^{(L,I)} \odot \sigma'' \left( \mathbf{z}^{(I)} \right) \right) \mathbf{B}^{(I)}$$

*where $\mathbf{B}^{(I)}$ is the Jacobian of $\mathbf{z}^{(I)}$ with respect to $\mathbf{x}$ (dimensions $N_I \times D$), and $\mathbf{F}^{(L,I)}$ is the Jacobian of $\mathbf{z}^{(L)}$ with respect to $\mathbf{a}^{(I)}$ (dimensions $N_L \times N_I$).*

The proof is presented in Appendix D.3. Using the chain rule of gradient, we can compute $\mathbf{B}^{(I)}$, $\mathbf{F}^{(L,I)}$ matrices in Lemma 1 recursively as follows:

$$\mathbf{B}^{(1)} = \mathbf{W}^{(1)} \qquad \mathbf{B}^{(I)} = \mathbf{W}^{(I)} diag \left( \sigma' \left( \mathbf{z}^{(I-1)} \right) \right) \mathbf{B}^{(I-1)} \quad I \in [2, L-1] \quad (4)$$

$$\mathbf{F}^{(L,L-1)} = \mathbf{W}^{(L)} \qquad \mathbf{F}^{(L,I)} = \mathbf{W}^{(L)} diag \left( \sigma' \left( \mathbf{z}^{(L-1)} \right) \right) \mathbf{F}^{(L-1,I)} \quad I \in [L-2] \quad (5)$$

This leads to a fast back-propagation like method that can be used to compute the Hessian. Note that Lemma 1 only assumes a matrix multiplication operation from $\mathbf{a}^{(I-1)}$ to $\mathbf{z}^{(I)}$. Since a convolution operation can also be expressed as a matrix multiplication, we can directly extend this lemma to deep convolutional networks. Furthermore, Lemma 1 can also be of independent interest in other related problems such as higher-order interpretation methods for deep learning (e.g. Singla et al. (2019)).

### 4.2 Curvature bounds for Two Layer networks

For a two-layer network and using Lemma 1, $\nabla_{\mathbf{x}}^2 \left( \mathbf{z}_y^{(2)} - \mathbf{z}_t^{(2)} \right)$ is given by:

$$\nabla_{\mathbf{x}}^2 \left( \mathbf{z}_y^{(2)} - \mathbf{z}_t^{(2)} \right) = \left( \mathbf{W}^{(1)} \right)^T diag \left( \left( \mathbf{W}_y^{(2)} - \mathbf{W}_t^{(2)} \right) \odot \sigma'' \left( \mathbf{z}^{(1)} \right) \right) \mathbf{W}^{(1)}$$

Note that only $\sigma''(\mathbf{z}^{(1)})$ depends on $\mathbf{x}$. We can maximize and minimize each element in the diag term, $(\mathbf{W}_{y,i}^{(2)} - \mathbf{W}_{t,i}^{(2)})\sigma''(\mathbf{z}_i^{(1)})$ independently subject to the constraint that $\sigma''(.)$ is bounded. Using this procedure, we construct matrices $\mathbf{P}$ and $\mathbf{N}$ that satisfy properties given in the following theorem:

**Theorem 3.** *Given a two layer network whose activation function has bounded second derivative:*

$$h_L \le \sigma''(x) \le h_U \quad \forall x \in \mathbb{R}$$

*(a) We have the following linear matrix inequalities (LMIs):*

$$\mathbf{N} \preccurlyeq \nabla_{\mathbf{x}}^2 \left( \mathbf{z}_y^{(2)} - \mathbf{z}_t^{(2)} \right) \preccurlyeq \mathbf{P} \qquad \forall \mathbf{x} \in \mathbb{R}^D$$

*(b) If $h_U \ge 0$ and $h_L \le 0$, $\mathbf{P}$ is a PSD matrix, $\mathbf{N}$ is a NSD matrix.*

*(c) This gives the following global bounds on the eigenvalues of the Hessian:*

$$m\mathbf{I} \preccurlyeq \nabla_{\mathbf{x}}^2 \left( \mathbf{z}_y^{(2)} - \mathbf{z}_t^{(2)} \right) \preccurlyeq M\mathbf{I}, \qquad \text{where } M = \lambda_{max}(\mathbf{P}), \; m = \lambda_{min}(\mathbf{N})$$

$\mathbf{P}$ *and* $\mathbf{N}$ *are independent of* $\mathbf{x}$ *and defined in equations* (55) *and* (56) *in Appendix D.4.*

The proof is presented in Appendix D.4. Because power iteration finds the eigenvalue with largest magnitude, we can use it to find $m$ and $M$ only when $\mathbf{P}$ is PSD and $\mathbf{N}$ is NSD. We solve for $h_U$ and $h_L$ for sigmoid, tanh, softplus activation functions in Appendix E and show that this is in fact the case for them. Note that this result does not hold for ReLU networks since the ReLU function is not differentiable. However, in Appendix $F$, we devise a method to compute the certificate for a two layer ReLU network by finding a quadratic lower bound for $\mathbf{z}_y^{(2)} - \mathbf{z}_t^{(2)}$.

### 4.3 CURVATURE BOUNDS FOR DEEP NETWORKS

Using Lemma 1, we know that $\nabla_{\mathbf{x}}^2 \mathbf{z}_i^{(L)}$ is a sum product of matrices $\mathbf{B}^{(I)}$ and $\mathbf{F}_i^{(L,I)}$. Thus, if we can find upper bounds for $\|\mathbf{B}^{(I)}\|$ and $\|\mathbf{F}_i^{(L,I)}\|$, we can get upper bounds for $\|\nabla_{\mathbf{x}}^2 \mathbf{z}_i^{(L)}\|$. Using this intuition (details are presented in Appendix D.5), we have the following result:

**Theorem 4.** *Given an $L$ layer neural network whose activation function satifies:*

$$|\sigma'(x)| \le g, \; |\sigma''(x)| \le h \qquad \forall x \in \mathbb{R},$$

*the absolute value of eigenvalues of* $\nabla_{\mathbf{x}}^2 \mathbf{z}_i^{(L)}$ *is globally bounded by the following quantity:*

$$\left\| \nabla_{\mathbf{x}}^2 \mathbf{z}_i^{(L)} \right\| \le h \sum_{I=1}^{L-1} \left( r^{(I)} \right)^2 \max_j \left( \mathbf{S}_{i,j}^{(L,I)} \right), \quad \forall \mathbf{x} \in \mathbb{R}^D$$

*where* $r^{(I)}$ *and* $\mathbf{S}^{(L,I)}$ *are independent of* $\mathbf{x}$ *and defined recursively as:*

$$r^{(1)} = \left\| \mathbf{W}^{(1)} \right\|, \qquad\qquad r^{(I)} = g \left\| \mathbf{W}^{(I)} \right\| r^{(I-1)} \quad I \in [2, L-1] \qquad (6)$$

$$\mathbf{S}^{(L,L-1)} = \left| \mathbf{W}^{(L)} \right|, \qquad\qquad \mathbf{S}^{(L,I)} = g \left| \mathbf{W}^{(L)} \right| \mathbf{S}^{(L-1,I)} \quad I \in [L-2] \qquad (7)$$

The above expressions allows for an efficient computation of the curvature bounds for deep networks. We consider simplification of this result for sigmoid, tanh, softplus activations in Appendix E.

Note that bounds for $\mathbf{z}_y^{(L)} - \mathbf{z}_t^{(L)}$ can be computed by replacing $\mathbf{W}_i^{(L)}$ with $\mathbf{W}_y^{(L)} - \mathbf{W}_t^{(L)}$ in Theorem 4. The resulting bound is independent of $\mathbf{x}$, and only depends on network weights $\mathbf{W}^{(I)}$, the true label $y$, and the target $t$. We denote it with $K(\mathbf{W}, y, t)$. To simplify notation, when no confusion arises we denote it with $K$. In our experiments, for two layer networks, we use $M, m$ from Theorem 3 (since it provides tighter curvature bounds). For deeper networks ($L \ge 3$), we use $M = K$, $m = -K$.

## 5 ADVERSARIAL TRAINING WITH CURVATURE REGULARIZATION

Using Theorem 2 (b), we know that if we solve the curvature-based attack optimization and obtain $\rho = \|\mathbf{x}^{(attack)} - \mathbf{x}^{(0)}\|$, $\mathbf{x}^{(attack)}$ is provably the closest adversarial example to $\mathbf{x}^{(0)}$. However, when

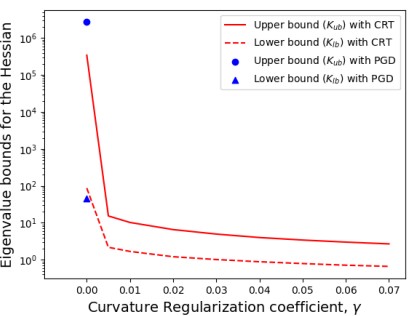

Figure 1: $K_{ub}$ and $K_{lb}$ are upper and lower curvature bounds of the network with Sigmoid activations (averaged over $(y, t)$ pairs). When $\gamma = 0$ (no curvature regularization), networks adversarially trained with CRT or PGD both have high curvatures. However, CRT even with a small $\gamma$ leads to a significant decrease in curvature bounds (note the log-scale of $y$-axis). Similar results hold for networks with Tanh activations (Appendix Figure 2)

we performed adversarial training (with $\rho = 0.5$), we found that the curvature bound is loose and almost none of training inputs lead to zero primal-dual gap with $\rho = \|\mathbf{x}^{(attack)} - \mathbf{x}^{(0)}\|$. To fix this issue, we use a regularizer that penalizes the curvature bound, $K$. Using equations (6) and (7), we can compute $K$ using absolute value, matrix multiplications, and operator norm $\left(\|\mathbf{W}^{(I)}\|, I \in [L]\right)$. Since the gradient of operator norm does not exist in standard libraries, we created a new layer where the gradient of $\|\mathbf{W}^{(I)}\|$, i.e $\nabla_{\mathbf{W}^{(I)}}\|\mathbf{W}^{(I)}\|$ is given by:

$$\nabla_{\mathbf{W}^{(I)}}\|\mathbf{W}^{(I)}\| = \mathbf{u}^{(I)}\left(\mathbf{v}^{(I)}\right)^T \qquad \mathbf{u}^{(I)}, \mathbf{v}^{(I)} \text{ satisfy } \mathbf{W}^{(I)}\mathbf{v}^{(I)} = \|\mathbf{W}^{(I)}\|\mathbf{u}^{(I)}$$

This approach to compute the gradient of the largest singular value of a matrix has also been used in previous ICLR work (Miyato et al., 2018). Implementation details are in Appendix C.1.

Thus, the per-sample loss for training with curvature regularization is:

$$cross\_entropy\left(\mathbf{z}^{(L)}(\mathbf{x}^{(0)}), \; y\right) + \gamma K(\mathbf{W}, y, t)$$

where $y$ is the true label of the input $\mathbf{x}^{(0)}$, $t$ is the target label and $\gamma$ is the regularizer for penalizing large curvatures. Similar to the adversarial training, in CRT, we use $\mathbf{x}^{(attack)}$ instead of $\mathbf{x}^{(0)}$.

## 6 EXPERIMENTS

The *empirical robust accuracy* means the fraction of test samples that were correctly classified after running an $l_2$ bounded PGD attack (Madry et al., 2018), the *certified robust accuracy* means the fraction of correctly classified test samples whose robustness certificates are greater than a pre-specified radius $\rho$. Unless otherwise specified, we use the class with the second largest logit as the attack target (i.e. the class $t$) and $\rho = 0.5$. All experiments were run on the MNIST dataset. The notation ($L \times [1024]$, activation) denotes a neural network with $L$ layers with the specified activation, ($\gamma = c$) denotes standard training with $\gamma$ set to $c$, while (CRT, $c$) denotes CRT training with $\gamma = c$. Certificates are computed over 150 randomly chosen correctly classified images.

**Comparison with existing certificates**: In Table 3, we compare CRC with CROWN-general (Zhang et al., 2018a). For 2-layer networks, CRC outperforms CROWN significantly. For deeper networks, CRC works better only when the network is trained with curvature regularization. However, even with small $\gamma = 0.005$, we see a significant increase in CRC but a very small drop in the test accuracy (without any adversarial training). We can see that with $\gamma = 0.01$, non-trivial certified accuracies of $83.53\%$, $88.33\%$, $89.61\%$ can be achieved on $2, 3, 4$ layer sigmoid networks, respectively, without any adversarial training. Adversarial training using CRT further boosts certified accuracy to $95.59\%$, $94.99\%$ and $93.41\%$, respectively. We observe similar results with Tanh networks in Appendix Section G.2. We show some results on CIFAR-10 dataset in Table 6. We again observe improvements in the robustness certificate and certified robust accuracy using CRC and CRT.

In Figure 1, we plot the effect of $\gamma$ on the curvature upper bound $K_{ub}$ and a lower bound $K_{lb}$ of a 4-layer network with Sigmoid activations. $K_{lb}$ is computed by taking the maximum of the largest eigenvalue of the Hessian across all test images with label $y$ and the second largest logit $t$, then averaging across different $(y, t)$. Similarly, $K_{ub}$ is the mean of $K$ over all pairs $(y, t)$ (details in Appendix G.5). We observe that without any curvature regularization (when $\gamma = 0$), both standard adversarial training with PGD as well as the CRT lead to networks with high curvatures. However,

CRT with even a small $\gamma$ leads to a significant decrease in curvature bounds. Similar trends can be observed for networks with Tanh activations (Appendix Figure 2). Curvature bounds are higher for the Tanh networks compared to the Sigmoid ones due to having larger $g$ and $h$ parameters for Tanh in Theorem 4. Moreover, we report curvature bounds for networks with different depth in Appendix Table 12. We observe that increasing depth increases curvature bounds.

**Comparison with existing adversarial training methods**: We compare CRT with adversarial training methods namely PGD (Madry et al., 2018) and TRADES (Zhang et al., 2019) in Table 4. We observe that none of the other methods give higher certified accuracy or robustness certificates than our proposed methods. We observe similar results with Tanh networks (Appendix Table 9). Moreover, in Appendix Table 10, we observe that CRT outperforms Randomized Smoothing (Cohen et al., 2019) for 2 and 3 layer networks. Since TRADES and Randomized Smoothing were designed for untargeted attacks while CRT is for targeted attacks, to have a fair comparison, we modify the multi-class version of the cross entropy loss with its binary version (details in Appendix Section G.3). However, we emphasize that there is a fundamental difference between our certificate (and CROWN) with the smoothing based method since our certificate is deterministic while the smoothing-based certificate is probabilistic (with high probability).

We also compare against robust training technique proposed in Wong et al. (2018) in Table 5 with $l_2$ radius $\rho = 1.58$. We note that the activation functions used in this comparison are different because the method proposed in Wong et al. (2018) uses ReLU while our proposed CRT can only work with fully differentiable activation functions such as softplus, sigmoid, tanh. For CRT, we use a network with softplus activation function with curvature regularization. We observe that for two and three layer networks, CRT gives higher certified robust accuracy compared to that of Wong et al. (2018) while the difference between certified accuracy of the two approaches for a 4 layer network is small (around $0.22\%$). We emphasize that to get good certified accuracy using our method, we need to add curvature regularization during the training. Otherwise, the curvature bounds and therefore the robustness certificates are loose.

| Network | Training | Standard Accuracy | Certified Robust Accuracy | Certificate (mean) | | Time per image (seconds) | |
|---|---|---|---|---|---|---|---|
| | | | | CROWN | CRC | CROWN | CRC |
| 2×[1024], sigmoid | standard | 98.37% | 54.17% | 0.28395 | **0.48500** | 0.1818 | 0.1911 |
| | $\gamma = 0.005$ | 97.96% | 82.68% | 0.36125 | **0.83367** | 0.1599 | 0.2229 |
| | $\gamma = 0.01$ | 98.08% | 83.53% | 0.32548 | **0.84719** | 0.1732 | 0.2186 |
| | CRT, 0.01 | 98.57% | **95.59%** | 0.43061 | **1.54673** | 0.1823 | 0.1910 |
| 3×[1024], sigmoid | standard | 98.37% | 0.00% | **0.24644** | 0.06874 | 1.6356 | 0.5012 |
| | $\gamma = 0.005$ | 97.98% | 88.66% | 0.38030 | **0.99044** | 1.6220 | 0.5319 |
| | $\gamma = 0.01$ | 97.71% | 88.33% | 0.39799 | **1.07842** | 1.6342 | 0.5295 |
| | CRT, 0.01 | 97.23% | **94.99%** | 0.39603 | **1.24100** | 1.5625 | 0.5013 |
| 4×[1024], sigmoid | standard | 98.39% | 0.00% | **0.19501** | 0.00454 | 4.7814 | 0.8107 |
| | $\gamma = 0.005$ | 97.74% | 88.95% | 0.36863 | **0.91840** | 5.1667 | 0.8567 |
| | $\gamma = 0.01$ | 97.41% | 89.61% | 0.40620 | **1.05323** | 4.6296 | 0.8328 |
| | CRT, 0.01 | 97.83% | **93.41%** | 0.40327 | **1.06208** | 4.1830 | 0.8088 |

Table 3: Comparison between CROWN-general (Zhang et al., 2018a) and CRC. Note that both CROWN and CRC are computed on CPU. However, running time numbers are not directly comparable because our CRC implementation uses a batch of images while the CROWN implementation uses a single image at a time.

# 7 EXTENSIONS AND DISCUSSION

## 7.1 USING LOCAL INSTEAD OF GLOBAL CURVATURE BOUNDS

From Theorems 1 and 2, we can observe that if the curvature is *locally* bounded within a convex region around the input, then the corresponding dual problems ($d^*_{cert}$, $d^*_{attack}$) are convex optimization problems inside this region. This leads to the following result:

| Network | Training | Standard Accuracy | Empirical Robust Accuracy | Certified Robust Accuracy | Certificate (mean) | |
|---|---|---|---|---|---|---|
| | | | | | CROWN | CRC |
| 2×[1024], sigmoid | PGD | 98.80% | 96.26% | 93.37% | 0.37595 | 0.82702 |
| | TRADES | 98.87% | 96.76% | 95.13% | 0.41358 | 0.92300 |
| | CRT, 0.01 | 98.57% | 96.28% | **95.59%** | 0.43061 | **1.54673** |
| 3×[1024], sigmoid | PGD | 98.84% | 96.14% | 0.00% | 0.29632 | 0.07290 |
| | TRADES | 98.95% | 96.79% | 0.00% | 0.30576 | 0.09108 |
| | CRT, 0.01 | 98.23% | 95.70% | **94.99%** | 0.39603 | **1.24100** |
| 4×[1024], sigmoid | PGD | 98.84% | 96.26% | 0.00% | 0.25444 | 0.00658 |
| | TRADES | 98.76% | 96.67% | 0.00% | 0.26128 | 0.00625 |
| | CRT, 0.01 | 97.83% | 94.65% | **93.41%** | 0.40327 | **1.06208** |

Table 4: Comparison between CRT, PGD (Madry et al., 2018) and TRADES (Zhang et al., 2019).

| Network | Training | Standard Accuracy | Certified Robust Accuracy |
|---|---|---|---|
| 2×[1024], softplus | CRT, 0.01 | 98.68% | **69.79%** |
| 3×[1024], softplus | CRT, 0.01 | 98.26% | 14.21% |
| | CRT, 0.03 | 97.82% | 50.72% |
| | CRT, 0.05 | 97.43% | **57.78%** |
| 4×[1024], softplus | CRT, 0.01 | 97.80% | 6.25% |
| | CRT, 0.03 | 97.09% | 29.64% |
| | CRT, 0.05 | 96.33% | 44.44% |

| Network | Standard Accuracy | Certified Robust Accuracy |
|---|---|---|
| 2×[1024], relu | 89.33% | 44.29% |
| 3×[1024], relu | 89.12% | 44.21% |
| 4×[1024], relu | 90.17% | **44.66%** |

Table 5: Comparison between CRT (left table) and Convex Outer Adversarial Polytope (right table) (Wong et al., 2018) with attack radius $\rho = 1.58$.

| Network | Training | Standard Accuracy | Empirical Robust Accuracy | Certified Robust Accuracy | Certificate (mean) | |
|---|---|---|---|---|---|---|
| | | | | | CROWN | CRC |
| 2 × [1024], sigmoid | standard | 46.23% | 37.82% | 14.10% | 0.37219 | **0.38173** |
| | $\gamma = 0.01$ | 45.42% | 38.17% | 26.50% | 0.40540 | **0.55010** |
| 3 × [1024], sigmoid | standard | 48.57% | 34.80% | 0.00% | 0.19127 | 0.01404 |
| | $\gamma = 0.01$ | 50.31% | 39.87% | 18.28% | 0.24778 | **0.37895** |
| 4 × [1024], sigmoid | standard | 46.04% | 34.38% | 0.00% | 0.19340 | 0.00191 |
| | $\gamma = 0.01$ | 48.28% | 40.10% | 21.07% | 0.29654 | **0.40005** |

Table 6: Results for CIFAR-10 dataset (only curvature regularization, no CRT training)

**Corollary 1.** *Both Curvature-based Certificate and Attack Optimizations are convex optimization problems within an $l_2$ ball of radius $\rho$ around the input sample where curvature values are bounded.*

To use local curvature bounds, we need to ensure that the gradient descent trajectory does not escape the local region where curvature values are bounded. To do this, we reduce the step size by a factor of two until the trajectory lies inside the $l_2$ ball of radius $\rho$ around the input point.

It is straightforward to see that local bounds on curvature values can be tighter than global ones which can lead to better robustness certifications. In Table 7, we show significant improvements for the CRC certificate and some improvements on the certified accuracy for two-layer sigmoid and tanh networks on the MNIST dataset. In this case, to compute local curvature bounds, we need to have a bound on $h_L$ and $h_U$ (Theorem 3) within the ball of radius $\rho$ around the input sample. These bounds can be

significantly tighter than the global bounds. For example, if a sigmoid neuron's output is bounded by -0.4 and 0.6 (those bounds can be obtained efficiently using CROWN), the second derivative of sigmoid is bounded between -0.048 and +0.048, much better than the worst case bound -0.09623 and +0.09623 used in current global curvature bound.

| Network | Training | Certified Accuracy (Global) | Certified Accuracy (Local) | CRC Certificate (Global) | CRC Certificate (Local) |
|---|---|---|---|---|---|
| 2×[1024], sigmoid | CRT, 0.0 | 95.04% | 95.31% | 1.0011 | **1.3901** |
| | CRT, 0.01 | 95.59% | 95.59% | 1.5705 | **1.7262** |
| | CRT, 0.02 | 95.21% | 95.21% | 1.6720 | **1.7397** |
| 2×[1024], tanh | CRT, 0.0 | 92.69% | 94.39% | 0.8028 | **1.0511** |
| | CRT, 0.01 | 95.00% | 95.00% | 1.4832 | **1.6068** |
| | CRT, 0.02 | 94.77% | 94.77% | 1.5848 | **1.6592** |

Table 7: Comparison between Certified Robust accuracy and CRC for 2 layer sigmoid and tanh networks using global and local curvature bounds

For deeper networks, however, computing local curvature bounds is more challenging. We leave exploring this direction for the future work.

## 7.2 EXTENSION TO CONVOLUTIONAL NEURAL NETWORKS

The formula derived in Lemma 1 is valid even for convolutional neural networks. However, to use our methods, we need to bound the singular values of the Jacobian of the convolution operation. In order to do this, one can use spectral bounds for convolution layers derived in Sedghi et al. (2018). Furthermore, other heuristic techniques such as the one proposed in Miyato et al. (2018) can be used for the curvature regularization as well. We present some prelimiary results using these techniques for 2 layer convolutional networks on MNIST and CIFAR-10 in Appendix Section G.1.

## 8 CONCLUSION

In this paper, we develop computationally-efficient convex relaxations for robustness certification and adversarial attack problems given the classifier has a bounded curvature. We also show that this convex relaxation is tight under some general conditions. To be able to use proposed certification and attack convex optimizations, we derive global curvature bounds for deep networks with differentiable activation functions. This result is a consequence of a closed-form expression that we derived for the Hessian of a deep network. Our empirical results indicate that our proposed curvature-based robustness certificate outperforms the CROWN certificate by an order of magnitude while being faster to compute as well. Furthermore, adversarial training using our attack method coupled with curvature regularization results in a significantly higher certified robust accuracy than the existing adversarial training methods. Scaling up our proposed curvature-based robustness certification and training methods as well as further tightening the derived curvature bounds are among interesting directions for the future work.

## 9 ACKNOWLEDGMENT

We thank the reviewers for their feedback and comments specially on extensions of our results to characterize local curvature bounds.

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

# Appendix

## A  RELATED WORK

Many defenses have been proposed to make neural networks robust against adversarial examples. These methods can be classified into empirical defenses which empirically seem to be robust against known adversarial attacks, and certified defenses, which are provably robust against such attacks.

**Empirical defenses** The best known empirical defense is adversarial training (Kurakin et al., 2016b; Madry et al., 2018; Zhang et al., 2019). In this method, a neural network is trained to minimize the worst-case loss over a region around the input. Although such defenses seem to work on existing attacks, there is no guarantee that a more powerful attack would not break them. In fact, most such defenses proposed in the literature were later broken by stronger attacks (Athalye et al., 2018; Carlini & Wagner, 2017; Uesato et al., 2018; Athalye & Carlini, 2018). To end this arms race between defenses and attacks, a number of works have tried to focus on certified defenses that have formal robustness guarantees.

**Certified defenses** A classifier is said to be certifiably robust if one can easily obtain a guarantee that a classifier's prediction remains constant within some region around the input. Such defenses typically rely on certification methods which are either exact or conservative. Exact methods report whether or not there exists a adversarial perturbation inside some $l_p$ norm ball. In contrast, conservative methods either certify that no adversarial perturbation exists or decline to make a certification; they may decline even when no such perturbation exists. Exact methods are usually based on Satisfiability Modulo Theories (Huang et al., 2016; Katz et al., 2017; Ehlers, 2017; Carlini et al., 2017) and Mixed Integer linear programming (Cheng et al., 2017; Lomuscio & Maganti, 2017; Dutta et al., 2018; Fischetti & Jo, 2018; Bunel et al., 2017). Unfortunately, they are computationally inefficient and difficult to scale up to even moderately sized neural networks. In contrast, conservative methods are more scalable and efficient which makes them useful for building certified defenses (Wang et al., 2018a; Wong & Kolter, 2017; Wang et al., 2018b; Wong et al., 2018; Raghunathan et al., 2018b;a; Dvijotham et al., 2018a;b; Croce et al., 2018; Singh et al., 2018; Gowal et al., 2018; Gehr et al., 2018; Mirman et al., 2018; Zhang et al., 2018b; Weng et al., 2018). However, even these methods have not been shown to scale to practical networks that are large and expressive enough to perform well on ImageNet, for example. To scale to such large networks, randomized smoothing has been proposed as a *probabilistically* certified defense.

**Randomized smoothing** Randomized smoothing was previously proposed by several works (Liu et al., 2017; Cao & Gong, 2017) as a empirical defense without any formal guarantees. Lécuyer et al. (2018) first proved robustness guarantees for randomized smoothing classifier using inequalities from differential privacy. Li et al. (2018) improved upon the same using tools from information theory. Recently, Cohen et al. (2019) provided a even tighter robustness guarantee for randomized smoothing. Salman et al. (2019) proposed a method of adversarial training for the randomized smoothing classifier giving state of the art results in the $l_2$ norm metric.

## B  THE ATTACK PROBLEM

For a given input $\mathbf{x}^{(0)}$ with true label $y$ and attack target $t$, consider the attack problem. We are given that the eigenvalues of the Hessian $\nabla_{\mathbf{x}}^2 \left( \mathbf{z}_y^{(L)} - \mathbf{z}_t^{(L)} \right)$ are bounded below i.e:

$$m\mathbf{I} \preccurlyeq \nabla_{\mathbf{x}}^2 \left( \mathbf{z}_y^{(L)} - \mathbf{z}_t^{(L)} \right) \qquad \forall \mathbf{x} \in \mathbb{R}^D$$

Here $m < 0$ (since $\mathbf{z}_y^{(L)} - \mathbf{z}_t^{(L)}$ is not convex in general).

The goal here is to find an adversarial example inside a $l_2$ ball of radius $\rho$ such that $(\mathbf{z}_y^{(L)} - \mathbf{z}_t^{(L)})(\mathbf{x})$ is minimized. That is, we want to solve the following optimization:

$$p_{attack}^* = \min_{\|\mathbf{x} - \mathbf{x}^{(0)}\| \le \rho} \left[ \left( \mathbf{z}_y^{(L)} - \mathbf{z}_t^{(L)} \right)(\mathbf{x}) \right] = \min_{\mathbf{x}} \max_{\eta \ge 0} \left[ \left( \mathbf{z}_y^{(L)} - \mathbf{z}_t^{(L)} \right)(\mathbf{x}) + \frac{\eta}{2} \left( \left\| \mathbf{x} - \mathbf{x}^{(0)} \right\|^2 - \rho^2 \right) \right]$$

(8)

This optimization can be hard in general. Using the max-min inequality (primal $\geq$ dual), we have:

$$p^*_{attack} \geq \max_{\eta \geq 0} d_{attack}(\eta), \quad d_{attack}(\eta) = \min_{\mathbf{x}} \left[ \left(\mathbf{z}^{(L)}_y - \mathbf{z}^{(L)}_t\right)(\mathbf{x}) + \frac{\eta}{2}\left(\left\|\mathbf{x} - \mathbf{x}^{(0)}\right\|^2 - \rho^2\right) \right] \quad (9)$$

We know that for every $\eta \geq 0$, $d_{attack}(\eta)$ gives a lower bound to the primal solution $p^*_{attack}$. But solving $d_{attack}(\eta)$ for any $\eta \geq 0$ can be hard unless the objective is convex. We prove that if the eigenvalues of the Hessian are bounded below i.e:

$$m\mathbf{I} \preccurlyeq \nabla^2_{\mathbf{x}}\left(\mathbf{z}^{(L)}_y - \mathbf{z}^{(L)}_t\right) \qquad \forall \mathbf{x} \in \mathbb{R}^D$$

In general $m < 0$, since $(\mathbf{z}^{(L)}_y - \mathbf{z}^{(L)}_t)$ is non-convex.
$d_{attack}(\eta)$ is a convex optimization problem for $-m \leq \eta$. Equivalently the objective function, i.e the function inside the $\min_{\mathbf{x}}$:

$$\left[ \left(\mathbf{z}^{(L)}_y - \mathbf{z}^{(L)}_t\right)(\mathbf{x}) + \frac{\eta}{2}\left(\left\|\mathbf{x} - \mathbf{x}^{(0)}\right\|^2 - \rho^2\right) \right] \quad \text{is a convex function in } \mathbf{x} \text{ for } -m \leq \eta$$

The Hessian of the above function is given by:

$$\nabla^2_{\mathbf{x}}\left(\mathbf{z}^{(L)}_y - \mathbf{z}^{(L)}_t\right) + \eta\mathbf{I}$$

Since we know that eigenvalues of $\nabla^2_{\mathbf{x}}(\mathbf{z}^{(L)}_y - \mathbf{z}^{(L)}_t) \succcurlyeq m\mathbf{I}$, we know that eigenvalues of the above Hessian are $\geq \eta + m$. For $\eta \geq -m$, the eigenvalues are positive implying that the objective function is convex.

Since $d_{attack}(\eta)$ gives a lower bound to $p^*_{attack}$ for every $\eta \geq 0$, we get the following result:

$$p^*_{attack} \geq d^*_{attack} \text{ where } d^*_{attack} = \max_{-m \leq \eta} d_{attack}(\eta) \quad (10)$$

Note that if $\mathbf{x}^{(attack)}$ is the solution to $d^*_{attack}$ such that: $\left\|\mathbf{x}^{(attack)} - \mathbf{x}^{(0)}\right\| = \rho$, by the definition of $d^*_{attack}$:

$$d^*_{attack} = \left(\mathbf{z}^{(L)}_y - \mathbf{z}^{(L)}_t\right)(\mathbf{x}^{(attack)})$$

But then by the definition of $p^*_{attack}$, $p^*_{attack} \leq d^*_{attack}$, implying that the duality gap is zero, i.e $p^*_{attack} = d^*_{attack}$. This procedure leads to the theorem 2.

## C  IMPLEMENTATION DETAILS

### C.1  COMPUTING THE DERIVATIVE OF LARGEST SINGULAR VALUE

Our objective is to compute derivative of the largest singular value, i.e $\|\mathbf{W}^{(I)}\|$ with respect to $\mathbf{W}^{(I)}$.

$\mathbf{u}^{(I)}, \mathbf{v}^{(I)}$ are the singular vectors such that:

$$\mathbf{W}^{(I)}\mathbf{v}^{(I)} = \|\mathbf{W}^{(I)}\|\mathbf{u}^{(I)}$$

$\mathbf{v}^{(I)}$, $\|\mathbf{W}^{(I)}\|^2$ can be computed by running power iteration on $\left(\mathbf{W}^{(I)}\right)^T \mathbf{W}^{(I)}$. $\mathbf{u}^{(I)}$ can be computed using the identity:

$$\mathbf{u}^{(I)} = \frac{\mathbf{W}^{(I)}\mathbf{v}^{(I)}}{\gamma^{(I)}}$$

We use 25 iterations of the power method to compute the above quantities.

### C.2  UPDATE EQUATION FOR THE CERTIFICATE PROBLEM

Our goal is to minimize $\left\|\mathbf{x} - \mathbf{x}^{(0)}\right\|$ such that $\left(\mathbf{z}^{(L)}_y - \mathbf{z}^{(L)}_t\right)(\mathbf{x}) = 0$. We know that the Hessian satisfies the following LMIs:

$$m\mathbf{I} \preccurlyeq \nabla^2_{\mathbf{x}}\left(\mathbf{z}^{(L)}_y - \mathbf{z}^{(L)}_t\right) \preccurlyeq M\mathbf{I} \quad (11)$$

$K$ is given by Theorem 4 for neural network of any depth ($L \geq 2$). For 2 layer networks, $M$ and $m$ are given by Theorem 3. But for deeper networks ($L \geq 3$), $M = K$, $m = -K$. In either case, $K \geq \max(|m|, |M|)$. Thus, we also have:

$$-K\mathbf{I} \preccurlyeq \nabla^2_{\mathbf{x}}\left(\mathbf{z}^{(L)}_y - \mathbf{z}^{(L)}_t\right) \preccurlyeq K\mathbf{I} \tag{12}$$

We will solve the dual ($d^*_{cert}$) of the attack problem ($p^*_{cert}$).

The primal problem ($p^*_{cert}$) is given by:

$$p^*_{cert} = \min_{\mathbf{z}^{(L)}_y(\mathbf{x}) = \mathbf{z}^{(L)}_t(\mathbf{x})} \left[\frac{1}{2}\left\|\mathbf{x} - \mathbf{x}^{(0)}\right\|^2\right] = \min_{\mathbf{x}} \max_{\eta} \left[\frac{1}{2}\left\|\mathbf{x} - \mathbf{x}^{(0)}\right\|^2 + \eta\left(\mathbf{z}^{(L)}_y - \mathbf{z}^{(L)}_t\right)(\mathbf{x})\right]$$

Using inequality (11) and Theorem 1 part (a), we know that the dual of the above problem is convex when $-1/M \leq \eta \leq -1/m$.

The corresponding dual problem ($d^*_{cert}$) is given by:

$$d^*_{cert} = \max_{-1/M \leq \eta \leq -1/m} d_{cert}(\eta), \qquad d_{cert}(\eta) = \min_{\mathbf{x}}\left[\frac{1}{2}\left\|\mathbf{x} - \mathbf{x}^{(0)}\right\|^2 + \eta\left(\mathbf{z}^{(L)}_y - \mathbf{z}^{(L)}_t\right)(\mathbf{x})\right]$$

For a given $\eta$, we have the following optimization:

$$d_{cert}(\eta) = \min_{\mathbf{x}}\left[\frac{1}{2}\|\mathbf{x} - \mathbf{x}^{(0)}\|^2 + \eta\left(\mathbf{z}^{(L)}_y - \mathbf{z}^{(L)}_t\right)(\mathbf{x})\right]$$

We will use majorization-minimization to solve this optimization.

At a point $\mathbf{x}^{(k)}$, we aim to solve for the point $\mathbf{x}^{(k+1)}$ that decreases the objective function. Using the Taylor's theorem at point $\mathbf{x}^{(k)}$, we have:

$$\left(\mathbf{z}^{(L)}_y - \mathbf{z}^{(L)}_t\right)(\mathbf{x})$$
$$= \left(\mathbf{z}^{(L)}_y - \mathbf{z}^{(L)}_t\right)(\mathbf{x}^{(k)}) + \left(\mathbf{g}^{(k)}\right)^T\left(\mathbf{x} - \mathbf{x}^{(k)}\right) + \frac{1}{2}\left(\mathbf{x} - \mathbf{x}^{(k)}\right)^T\mathbf{H}^{(\xi)}\left(\mathbf{x} - \mathbf{x}^{(k)}\right)$$

where $\mathbf{g}^{(k)}$ is the gradient of $(\mathbf{z}^{(L)}_y - \mathbf{z}^{(L)}_t)$ at $\mathbf{x}^{(k)}$ and $\mathbf{H}^{(\xi)}$ is the Hessian at a point $\xi$ on the line connecting $\mathbf{x}$ and $\mathbf{x}^{(k)}$.

Multiplying both sides by $\eta$, we get the following equation:

$$\eta\left(\mathbf{z}^{(L)}_y - \mathbf{z}^{(L)}_t\right)(\mathbf{x})$$
$$= \eta\left(\mathbf{z}^{(L)}_y - \mathbf{z}^{(L)}_t\right)(\mathbf{x}^{(k)}) + \eta\left(\mathbf{g}^{(k)}\right)^T\left(\mathbf{x} - \mathbf{x}^{(k)}\right) + \frac{\eta}{2}\left(\mathbf{x} - \mathbf{x}^{(k)}\right)^T\mathbf{H}^{(\xi)}\left(\mathbf{x} - \mathbf{x}^{(k)}\right) \tag{13}$$

Using inequality (12), we know that $-K\mathbf{I} \preccurlyeq \mathbf{H}^{(\xi)} \preccurlyeq K\mathbf{I} \quad \forall \xi \in \mathbb{R}^D$,

$$\frac{\eta}{2}\left(\mathbf{x} - \mathbf{x}^{(k)}\right)^T\mathbf{H}^{(\xi)}\left(\mathbf{x} - \mathbf{x}^{(k)}\right) \leq \frac{|\eta K|}{2}\left\|\mathbf{x} - \mathbf{x}^{(k)}\right\|^2 \tag{14}$$

Using equation (13) and inequality (14):

$$\eta\left(\mathbf{z}^{(L)}_y - \mathbf{z}^{(L)}_t\right)(\mathbf{x}) \leq \eta\left(\mathbf{z}^{(L)}_y - \mathbf{z}^{(L)}_t\right)(\mathbf{x}^{(k)}) + \eta\left(\mathbf{g}^{(k)}\right)^T\left(\mathbf{x} - \mathbf{x}^{(k)}\right) + \frac{|\eta K|}{2}\left\|\mathbf{x} - \mathbf{x}^{(k)}\right\|^2$$

Adding $1/2\|\mathbf{x} - \mathbf{x}^{(0)}\|^2$ to both sides, we get the following inequality:

$$\frac{1}{2}\left\|\mathbf{x} - \mathbf{x}^{(0)}\right\|^2 + \eta\left(\mathbf{z}^{(L)}_y - \mathbf{z}^{(L)}_t\right)(\mathbf{x})$$
$$\leq \frac{1}{2}\left\|\mathbf{x} - \mathbf{x}^{(0)}\right\|^2 + \eta\left(\mathbf{z}^{(L)}_y - \mathbf{z}^{(L)}_t\right)(\mathbf{x}^{(k)}) + \eta\left(\mathbf{g}^{(k)}\right)^T\left(\mathbf{x} - \mathbf{x}^{(k)}\right) + \frac{|\eta K|}{2}\left\|\mathbf{x} - \mathbf{x}^{(k)}\right\|^2$$

LHS is the objective function of $d_{cert}(\eta)$ and RHS is an upper bound. In majorization-minimization, we minimize an upper bound on the objective function. Thus we set the gradient of RHS with respect to $\mathbf{x}$ to zero and solve for $\mathbf{x}$:

$$\nabla_{\mathbf{x}}\left[\frac{1}{2}\left\|\mathbf{x} - \mathbf{x}^{(0)}\right\|^2 + \eta\left(\mathbf{z}^{(L)}_y - \mathbf{z}^{(L)}_t\right)(\mathbf{x}^{(k)}) + \eta\left(\mathbf{g}^{(k)}\right)^T\left(\mathbf{x} - \mathbf{x}^{(k)}\right) + \frac{|\eta K|}{2}\left\|\mathbf{x} - \mathbf{x}^{(k)}\right\|^2\right] = 0$$

$$\mathbf{x} - \mathbf{x}^{(0)} + \eta \mathbf{g}^{(k)} + |\eta K| \left( \mathbf{x} - \mathbf{x}^{(k)} \right) = 0$$

$$(1 + |\eta K|) \mathbf{x} - \mathbf{x}^{(0)} + \eta \mathbf{g}^{(k)} - |\eta K| \mathbf{x}^{(k)} = 0$$

$$\mathbf{x} = -(1 + |\eta K|)^{-1} \left( \eta \mathbf{g}^{(k)} - |\eta K| \mathbf{x}^{(k)} - \mathbf{x}^{(0)} \right)$$

This gives the following iterative equation:

$$\mathbf{x}^{(k+1)} = -(1 + |\eta K|)^{-1} \left( \eta \mathbf{g}^{(k)} - |\eta K| \mathbf{x}^{(k)} - \mathbf{x}^{(0)} \right) \tag{15}$$

### C.3 UPDATE EQUATION FOR THE ATTACK PROBLEM

Our goal is to minimize $\mathbf{z}_y^{(L)} - \mathbf{z}_t^{(L)}$ within an $l_2$ ball of radius of $\rho$. We know that the Hessian satisfies the following LMIs:

$$m\mathbf{I} \preccurlyeq \nabla_{\mathbf{x}}^2 \left( \mathbf{z}_y^{(L)} - \mathbf{z}_t^{(L)} \right) \preccurlyeq M\mathbf{I} \tag{16}$$

$K$ is given by Theorem 4 for neural network of any depth ($L \geq 2$). For 2 layer networks, $M$ and $m$ are given by Theorem 3. But for deeper networks ($L \geq 3$), $M = K$, $m = -K$. In either case, $K \geq \max(|m|, |M|)$. Thus, we also have:

$$-K\mathbf{I} \preccurlyeq \nabla_{\mathbf{x}}^2 \left( \mathbf{z}_y^{(L)} - \mathbf{z}_t^{(L)} \right) \preccurlyeq K\mathbf{I} \tag{17}$$

We solve the dual ($d_{attack}^*$) of the attack problem ($p_{attack}^*$) for the given radius $\rho$.

The primal problem ($p_{attack}^*$) is given by:

$$p_{attack}^* = \min_{\|\mathbf{x} - \mathbf{x}^{(0)}\| \leq \rho} \mathbf{z}_y^{(L)} - \mathbf{z}_t^{(L)} = \min_{\mathbf{x}} \max_{\eta \geq 0} \left[ \mathbf{z}_y^{(L)} - \mathbf{z}_t^{(L)} + \frac{\eta}{2} \left( \left\| \mathbf{x} - \mathbf{x}^{(0)} \right\|^2 - \rho^2 \right) \right]$$

Using inequality (16) and Theorem 2 part (a), we know that the dual of the above problem is convex when $-m \leq \eta$.

The corresponding dual problem ($d_{cert}^*$) is given by:

$$d_{attack}^* = \max_{\eta \geq -m} d_{attack}(\eta), \quad d_{attack}(\eta) = \min_{\mathbf{x}} \left[ \left( \mathbf{z}_y^{(L)} - \mathbf{z}_t^{(L)} \right) (\mathbf{x}) + \frac{\eta}{2} \left( \left\| \mathbf{x} - \mathbf{x}^{(0)} \right\|^2 - \rho^2 \right) \right]$$

For a given $\eta$, we have the following optimization:

$$d_{attack}(\eta) = \min_{\mathbf{x}} \left[ \left( \mathbf{z}_y^{(L)} - \mathbf{z}_t^{(L)} \right) (\mathbf{x}) + \frac{\eta}{2} \left( \left\| \mathbf{x} - \mathbf{x}^{(0)} \right\|^2 - \rho^2 \right) \right]$$

We will use majorization-minimization to solve this optimization.

At a point $\mathbf{x}^{(k)}$, we have to solve for the point $\mathbf{x}^{(k+1)}$ that decreases the objective function. Using the Taylor's theorem at point $\mathbf{x}^{(k)}$, we have:

$$\left( \mathbf{z}_y^{(L)} - \mathbf{z}_t^{(L)} \right) (\mathbf{x})$$
$$= \left( \mathbf{z}_y^{(L)} - \mathbf{z}_t^{(L)} \right) \left( \mathbf{x}^{(k)} \right) + \left( \mathbf{g}^{(k)} \right)^T \left( \mathbf{x} - \mathbf{x}^{(k)} \right) + \frac{1}{2} \left( \mathbf{x} - \mathbf{x}^{(k)} \right)^T \mathbf{H}^{(\xi)} \left( \mathbf{x} - \mathbf{x}^{(k)} \right) \tag{18}$$

where $\mathbf{g}^{(k)}$ is the gradient of $(\mathbf{z}_y^{(L)} - \mathbf{z}_t^{(L)})$ at $\mathbf{x}^{(k)}$ and $\mathbf{H}^{(\xi)}$ is the Hessian at a point $\xi$ on the line connecting $\mathbf{x}$ and $\mathbf{x}^{(k)}$.

Using inequality 17, we know that $-K\mathbf{I} \preccurlyeq \mathbf{H}^{(\xi)} \preccurlyeq K\mathbf{I} \quad \forall \xi \in \mathbb{R}^D$,

$$\frac{1}{2} \left( \mathbf{x} - \mathbf{x}^{(k)} \right)^T \mathbf{H}^{(\xi)} \left( \mathbf{x} - \mathbf{x}^{(k)} \right) \leq \frac{K}{2} \left\| \mathbf{x} - \mathbf{x}^{(k)} \right\|^2 \tag{19}$$

Using equation (18) and inequality (19):

$$\left( \mathbf{z}_y^{(L)} - \mathbf{z}_t^{(L)} \right) (\mathbf{x}) \leq \left( \mathbf{z}_y^{(L)} - \mathbf{z}_t^{(L)} \right) (\mathbf{x}^{(k)}) + \left( \mathbf{g}^{(k)} \right)^T \left( \mathbf{x} - \mathbf{x}^{(k)} \right) + \frac{K}{2} \left\| \mathbf{x} - \mathbf{x}^{(k)} \right\|^2$$

Adding $\eta/2(\|\mathbf{x} - \mathbf{x}^{(0)}\|^2 - \rho^2)$ to both sides, we get the following inequality:

$$\left(\mathbf{z}_y^{(L)} - \mathbf{z}_t^{(L)}\right)(\mathbf{x}) + \frac{\eta}{2}\left(\|\mathbf{x} - \mathbf{x}^{(0)}\|^2 - \rho^2\right)$$

$$\leq \left(\mathbf{z}_y^{(L)} - \mathbf{z}_t^{(L)}\right)(\mathbf{x}^{(k)}) + \left(\mathbf{g}^{(k)}\right)^T\left(\mathbf{x} - \mathbf{x}^{(k)}\right) + \frac{K}{2}\|\mathbf{x} - \mathbf{x}^{(k)}\|^2 + \frac{\eta}{2}\left(\|\mathbf{x} - \mathbf{x}^{(0)}\|^2 - \rho^2\right)$$

LHS is the objective function of $d_{attack}(\eta)$ and RHS is an upper bound. In majorization-minimization, we minimize an upper bound on the objective function. Thus we set the gradient of RHS with respect to $\mathbf{x}$ to zero and solve for $\mathbf{x}$:

$$\nabla_{\mathbf{x}}\left[\left(\mathbf{z}_y^{(L)} - \mathbf{z}_t^{(L)}\right)(\mathbf{x}^{(k)}) + \left(\mathbf{g}^{(k)}\right)^T\left(\mathbf{x} - \mathbf{x}^{(k)}\right) + \frac{K}{2}\|\mathbf{x} - \mathbf{x}^{(k)}\|^2 + \frac{\eta}{2}\left(\|\mathbf{x} - \mathbf{x}^{(0)}\|^2 - \rho^2\right)\right] = 0$$

$$\mathbf{g}^{(k)} + K\left(\mathbf{x} - \mathbf{x}^{(k)}\right) + \eta\left(\mathbf{x} - \mathbf{x}^{(0)}\right) = 0$$

$$(K + \eta)\mathbf{x} + \mathbf{g}^{(k)} - K\mathbf{x}^{(k)} - \eta\mathbf{x}^{(0)} = 0$$

$$\mathbf{x} = -(K + \eta)^{-1}\left(\mathbf{g}^{(k)} - K\mathbf{x}^{(k)} - \eta\mathbf{x}^{(0)}\right)$$

This gives the following iterative equation:

$$\mathbf{x}^{(k+1)} = -(K + \eta)^{-1}\left(\mathbf{g}^{(k)} - K\mathbf{x}^{(k)} - \eta\mathbf{x}^{(0)}\right) \tag{20}$$

## C.4 Algorithm to compute the certificate

We start with the following initial values of $\mathbf{x}$, $\eta$, $\eta_{min}$, $\eta_{max}$:

$$\eta_{min} = -1/M, \qquad \eta_{max} = -1/m$$
$$\eta = \frac{1}{2}(\eta_{min} + \eta_{max}), \qquad \mathbf{x} = \mathbf{x}^{(0)}$$

To solve the dual for a given value of $\eta$, we run 20 iterations of the following update (derived in Appendix C.2):

$$\mathbf{x}^{(k+1)} = -(1 + |\eta K|)^{-1}\left(\eta\mathbf{g}^{(k)} - |\eta K|\mathbf{x}^{(k)} - \mathbf{x}^{(0)}\right)$$

To maximize the dual $d_{cert}(\eta)$ over $\eta$ in the range $[-1/M, -1/m]$, we use a bisection method: If the solution $\mathbf{x}$ for a given value of $\eta$, $(\mathbf{z}_y^{(L)} - \mathbf{z}_t^{(L)})(\mathbf{x}) > 0$, set $\eta_{min} = \eta$, else set $\eta_{max} = \eta$. Set the new $\eta = (\eta_{min} + \eta_{max})/2$ and repeat. The maximum number of updates to $\eta$ are set to 30. The routine to compute the certificate example is given in Algorithm 1.

## C.5 Algorithm to compute the attack

We start with the following initial values of $\mathbf{x}$, $\eta$, $\eta_{min}$, $\eta_{max}$:

$$\eta_{min} = -m, \qquad \eta_{max} = 20(1 - m)$$
$$\eta = \frac{1}{2}(\eta_{min} + \eta_{max}), \qquad \mathbf{x} = \mathbf{x}^{(0)}$$

To solve the dual for a given value of $\eta$, we run 20 iterations of the following update (derived in Appendix C.3):

$$\mathbf{x}^{(k+1)} = -(K + \eta)^{-1}\left(\mathbf{g}^{(k)} - K\mathbf{x}^{(k)} - \eta\mathbf{x}^{(0)}\right)$$

To maximize the dual $d_{cert}(\eta)$ over $\eta$ in the range $[-m, 20(1 - m)]$, we use a bisection method: If the solution $\mathbf{x}$ for a given value of $\eta$, $\|\mathbf{x} - \mathbf{x}^{(0)}\| \leq \rho$, set $\eta_{max} = \eta$, else set $\eta_{min} = \eta$. Set new $\eta = (\eta_{min} + \eta_{max})/2$ and repeat. The maximum number of updates to $\eta$ are set to 30. The routine to compute the attack example is given in Algorithm 2.

---

**Algorithm 1** Certificate optimization

---

**Require:** input $\mathbf{x}^{(0)}$, label $y$, target $t$
  $m, M, K \leftarrow compute\_bounds(\mathbf{z}_y^{(L)} - \mathbf{z}_t^{(L)})$
  $\eta_{min} \leftarrow -1/M$
  $\eta_{max} \leftarrow -1/m$
  $\eta \leftarrow 1/2(\eta_{min} + \eta_{max})$
  $\mathbf{x} \leftarrow \mathbf{x}^{(0)}$
  **for** $i$ in $[1, \dots, 30]$ **do**
    **for** $j$ in $[1, \dots, 20]$ **do**
      $\mathbf{g} \leftarrow compute\_gradient(\mathbf{z}_y^{(L)} - \mathbf{z}_t^{(L)}, \mathbf{x})$
      **if** $\|\eta\mathbf{g} + (\mathbf{x} - \mathbf{x}^{(0)})\| < 10^{-5}$ **then**
        **break**
      **end if**
      $\mathbf{x} \leftarrow -(1 + |\eta K|)^{-1}\left(\eta\mathbf{g} - |\eta K|\mathbf{x} - \mathbf{x}^{(0)}\right)$
    **end for**
    **if** $(\mathbf{z}_y^{(L)} - \mathbf{z}_t^{(L)})(\mathbf{x}) > 0$ **then**
      $\eta_{min} \leftarrow \eta$
    **else**
      $\eta_{max} \leftarrow \eta$
    **end if**
    $\eta \leftarrow (\eta_{min} + \eta_{max})/2$
  **end for**
  **return** $\mathbf{x}$

---

**Algorithm 2** Attack optimization

---

**Require:** input $\mathbf{x}^{(0)}$, label $y$, target $t$ , radius $\rho$
  $m, M, K \leftarrow compute\_bounds(\mathbf{z}_y^{(L)} - \mathbf{z}_t^{(L)})$
  $\eta_{min} \leftarrow -m$
  $\eta_{max} \leftarrow 20(1 - m)$
  $\eta \leftarrow 1/2(\eta_{min} + \eta_{max})$
  $\mathbf{x} \leftarrow \mathbf{x}^{(0)}$
  **for** $i$ in $[1, \dots, 30]$ **do**
    **for** $j$ in $[1, \dots, 20]$ **do**
      $\mathbf{g} \leftarrow compute\_gradient(\mathbf{z}_y^{(L)} - \mathbf{z}_t^{(L)}, \mathbf{x})$
      **if** $\|\mathbf{g} + \eta(\mathbf{x} - \mathbf{x}^{(0)})\| < 10^{-5}$ **then**
        **break**
      **end if**
      $\mathbf{x} \leftarrow -(K + \eta)^{-1}\left(\mathbf{g} - K\mathbf{x} - \eta\mathbf{x}^{(0)}\right)$
    **end for**
    **if** $\|\mathbf{x} - \mathbf{x}^{(0)}\| < \rho$ **then**
      $\eta_{max} \leftarrow \eta$
    **else**
      $\eta_{min} \leftarrow \eta$
    **end if**
    $\eta \leftarrow (\eta_{min} + \eta_{max})/2$
  **end for**
  **return** $\mathbf{x}$

---

# D  PROOFS

## D.1  PROOF OF THEOREM 1

(a)

$$d_{cert}(\eta) = \min_{\mathbf{x}} \left[ \frac{1}{2} \left\| \mathbf{x} - \mathbf{x}^{(0)} \right\|^2 + \eta \left( \mathbf{z}_y^{(L)}(\mathbf{x}) - \mathbf{z}_t^{(L)}(\mathbf{x}) \right) \right]$$

$$\nabla_{\mathbf{x}}^2 \left[ \frac{1}{2} \left\| \mathbf{x} - \mathbf{x}^{(0)} \right\|^2 + \eta \left( \mathbf{z}_y^{(L)}(\mathbf{x}) - \mathbf{z}_t^{(L)}(\mathbf{x}) \right) \right] = \mathbf{I} + \eta \nabla_{\mathbf{x}}^2 \left( \mathbf{z}_y^{(L)} - \mathbf{z}_t^{(L)} \right)$$

We are given that the Hessian $\nabla_{\mathbf{x}}^2 (\mathbf{z}_y^{(L)} - \mathbf{z}_t^{(L)})$ satisfies the following LMIs:

$$m\mathbf{I} \preccurlyeq \nabla_{\mathbf{x}}^2 \left( \mathbf{z}_y^{(L)} - \mathbf{z}_t^{(L)} \right) \preccurlyeq M\mathbf{I} \qquad \forall \mathbf{x} \in \mathbb{R}^n$$

The eigenvalues of $\mathbf{I} + \eta \nabla_{\mathbf{x}}^2 (\mathbf{z}_y^{(L)} - \mathbf{z}_t^{(L)})$ are bounded between:

$$(1 + \eta M, \ 1 + \eta m), \text{ if } \eta < 0$$
$$(1 + \eta m, \ 1 + \eta M), \text{ if } \eta > 0$$

We are given that $\eta$ satisfies the following inequalities where $m < 0, M > 0$ since $(\mathbf{z}_y^{(L)} - \mathbf{z}_t^{(L)})$ is neither convex, nor concave as a function of $\mathbf{x}$:

$$\frac{-1}{M} \le \eta \le \frac{-1}{m}, \qquad m < 0, M > 0$$

We have the following inequalities:

$$1 + \eta M \ge 0, \ 1 + \eta m \ge 0$$

Thus, $\mathbf{I} + \eta \nabla_{\mathbf{x}}^2 (\mathbf{z}_y^{(L)} - \mathbf{z}_t^{(L)})$ is a PSD matrix for all $\mathbf{x} \in \mathbb{R}^D$ when $-1/M \le \eta \le -1/m$ .
Thus $1/2 \|\mathbf{x} - \mathbf{x}^{(0)}\|^2 + \eta (\mathbf{z}_y^{(L)} - \mathbf{z}_t^{(L)})(\mathbf{x})$ is a convex function in $\mathbf{x}$ and $d_{cert}(\eta)$ is a convex optimization problem.

(b) For every value of $\eta$, $d_{cert}(\eta)$ is a lower bound for $p_{cert}^*$. Thus $d_{cert}^* = \max_{-1/M \le \eta \le -1/m} d_{cert}(\eta)$ is a lower bound for $p_{cert}^*$, i.e:

$$d_{cert}^* \le p_{cert}^* \tag{21}$$

Let $\eta^{(cert)}, \mathbf{x}^{(cert)}$ be the solution of the above dual optimization ($d_{cert}^*$) such that

$$\mathbf{z}_y^{(L)}(\mathbf{x}^{(cert)}) = \mathbf{z}_t^{(L)}(\mathbf{x}^{(cert)}) \tag{22}$$

$d_{cert}^*$ is given by the following:

$$d_{cert}^* = \frac{1}{2} \left\| \mathbf{x}^{(cert)} - \mathbf{x}^{(0)} \right\|^2 + \eta^{(cert)} \underbrace{\left( \mathbf{z}_y^{(L)}(\mathbf{x}^{(cert)}) - \mathbf{z}_t^{(L)}(\mathbf{x}^{(cert)}) \right)}_{=0}$$

Since we are given that $\mathbf{z}_y^{(L)}(\mathbf{x}^{(cert)}) = \mathbf{z}_t^{(L)}(\mathbf{x}^{(cert)})$, we get the following equation for $d_{cert}^*$:

$$d_{cert}^* = \frac{1}{2} \left\| \mathbf{x}^{(cert)} - \mathbf{x}^{(0)} \right\|^2 \tag{23}$$

Since $p_{cert}^*$ is given by the following equation:

$$p_{cert}^* = \min_{\mathbf{z}_y^{(L)}(\mathbf{x}) = \mathbf{z}_t^{(L)}(\mathbf{x})} \left[ \frac{1}{2} \left\| \mathbf{x} - \mathbf{x}^{(0)} \right\|^2 \right] \tag{24}$$

Using equations (22) and (24), $p_{cert}^*$ is the minimum value of $1/2 \|\mathbf{x} - \mathbf{x}^{(0)}\|^2 \quad \forall \mathbf{x} :$ $\mathbf{z}_y^{(L)}(\mathbf{x}) = \mathbf{z}_t^{(L)}(\mathbf{x})$:

$$p_{cert}^* \le \frac{1}{2} \left\| \mathbf{x}^{(cert)} - \mathbf{x}^{(0)} \right\|^2 \tag{25}$$

From equation (23), we know that $d_{cert}^* = 1/2 \|\mathbf{x}^{(cert)} - \mathbf{x}^{(0)}\|^2$. Thus, we get:

$$p_{cert}^* \le d_{cert}^* \tag{26}$$

Using equation (21) we have $d_{cert}^* \le p_{cert}^*$ and using (26), $p_{cert}^* \le d_{cert}^*$

$$p_{cert}^* = d_{cert}^*$$

## D.2 Proof of Theorem 2

(a)

$$d_{attack}(\eta) = \min_{\mathbf{x}} \left[ \left( \mathbf{z}_y^{(L)} - \mathbf{z}_t^{(L)} \right)(\mathbf{x}) + \frac{\eta}{2} \left( \left\| \mathbf{x} - \mathbf{x}^{(0)} \right\|^2 - \rho^2 \right) \right]$$

$$\nabla_{\mathbf{x}}^2 \left[ \left( \mathbf{z}_y^{(L)} - \mathbf{z}_t^{(L)} \right)(\mathbf{x}) + \frac{\eta}{2} \left\| \mathbf{x} - \mathbf{x}^{(0)} \right\|^2 \right] = \nabla_{\mathbf{x}}^2 \left( \mathbf{z}_y^{(L)} - \mathbf{z}_t^{(L)} \right) + \eta \mathbf{I}$$

Since the Hessian $\nabla_{\mathbf{x}}^2 (\mathbf{z}_y^{(L)} - \mathbf{z}_t^{(L)})$ is bounded below:

$$m\mathbf{I} \preccurlyeq \nabla_{\mathbf{x}}^2 \left( \mathbf{z}_y^{(L)} - \mathbf{z}_t^{(L)} \right) \qquad \forall \mathbf{x} \in \mathbb{R}^n$$

The eigenvalues of $\nabla_{\mathbf{x}}^2 (\mathbf{z}_y^{(L)} - \mathbf{z}_t^{(L)}) + \eta \mathbf{I}$ are bounded below:

$$(m + \eta)\mathbf{I} \preccurlyeq \nabla_{\mathbf{x}}^2 \left( \mathbf{z}_y^{(L)} - \mathbf{z}_t^{(L)} \right) + \eta \mathbf{I}$$

Since $\eta \geq -m$.

$$\eta + m \geq 0$$

Thus $\nabla_{\mathbf{x}}^2 (\mathbf{z}_y^{(L)} - \mathbf{z}_t^{(L)}) + \eta \mathbf{I}$ is a PSD matrix for all $\mathbf{x} \in \mathbb{R}^D$ when $\eta \geq -m$.
Thus $(\mathbf{z}_y^{(L)} - \mathbf{z}_t^{(L)})(\mathbf{x}) + \eta/2(\| \mathbf{x} - \mathbf{x}^{(0)} \|^2 - \rho^2)$ is a convex function in $\mathbf{x}$ and $d_{attack}(\eta)$ is a convex optimization problem.

(b) For every value of $\eta$, $d_{attack}(\eta)$ is a lower bound for $p_{attack}^*$. Thus $d_{attack}^* = \max_{-m \leq \eta} d_{attack}(\eta)$ is a lower bound for $p_{attack}^*$:

$$d_{attack}^* \leq p_{attack}^* \tag{27}$$

Let $\eta^{(attack)}, \mathbf{x}^{(attack)}$ be the solution of the above dual optimization $(d_{attack}^*)$ such that

$$\left\| \mathbf{x}^{(attack)} - \mathbf{x}^{(0)} \right\| = \rho \tag{28}$$

$d_{attack}^*$ is given by the following:

$$d_{attack}^* = \left( \mathbf{z}_y^{(L)} - \mathbf{z}_t^{(L)} \right)(\mathbf{x}^{(attack)}) + \frac{\eta^{(attack)}}{2} \underbrace{\left( \left\| \mathbf{x}^{(attack)} - \mathbf{x}^{(0)} \right\|^2 - \rho^2 \right)}_{=0}$$

Since we are given that $\| \mathbf{x}^{(attack)} - \mathbf{x}^{(0)} \| = \rho$, we get the following equation for $d_{attack}^*$:

$$d_{attack}^* = \left( \mathbf{z}_y^{(L)} - \mathbf{z}_t^{(L)} \right)(\mathbf{x}^{(attack)}) \tag{29}$$

Since $p_{attack}^*$ is given by the following equation:

$$p_{attack}^* = \min_{\| \mathbf{x} - \mathbf{x}^{(0)} \| \leq \rho} \left[ \left( \mathbf{z}_y^{(L)} - \mathbf{z}_t^{(L)} \right)(\mathbf{x}) \right] \tag{30}$$

Using equations (28) and (30), $p_{attack}^*$ is the minimum value of $(\mathbf{z}_y^{(L)} - \mathbf{z}_t^{(L)})(\mathbf{x}) \quad \forall \| \mathbf{x} - \mathbf{x}^{(0)} \| \leq \rho$:

$$p_{attack}^* \leq \left( \mathbf{z}_y^{(L)} - \mathbf{z}_t^{(L)} \right)(\mathbf{x}^{(attack)}) \tag{31}$$

From equation (29), we know that $d_{attack}^* = (\mathbf{z}_y^{(L)} - \mathbf{z}_t^{(L)})(\mathbf{x}^{(attack)})$. Thus, we get:

$$p_{attack}^* \leq d_{attack}^* \tag{32}$$

Using equation (27) we have $d_{attack}^* \leq p_{attack}^*$ and using (32), $p_{attack}^* \leq d_{attack}^*$

$$p_{attack}^* = d_{attack}^*$$

### D.3    PROOF OF LEMMA 1

We have to prove that for an $L$ layer neural network, the hessian of the $i^{th}$ hidden unit in the $L^{th}$ layer with respect to the input $\mathbf{x}$, i.e $\nabla_{\mathbf{x}}^2 \mathbf{z}_i^{(L)}$ is given by the following formula:

$$\nabla_{\mathbf{x}}^2 \mathbf{z}_i^{(L)} = \sum_{I=1}^{L-1} \left(\mathbf{B}^{(I)}\right)^T diag\left(\mathbf{F}_i^{(L,I)} \odot \sigma''\left(\mathbf{z}^{(I)}\right)\right)\mathbf{B}^{(I)} \tag{33}$$

where $\mathbf{B}^{(I)}$, $I \in [L]$ is a matrix of size $N_I \times D$ defined as follows:

$$\mathbf{B}^{(I)} = \left[\nabla_{\mathbf{x}}\mathbf{z}_1^{(I)}, \nabla_{\mathbf{x}}\mathbf{z}_2^{(I)}, \ldots, \nabla_{\mathbf{x}}\mathbf{z}_{N_I}^{(I)}\right]^T, \qquad I \in [L] \tag{34}$$

and $\mathbf{F}^{(L,I)}$, $I \in [L-1]$ is a matrix of size $N_L \times N_I$ defined as follows:

$$\mathbf{F}^{(L,I)} = \left[\nabla_{\mathbf{a}^{(I)}}\mathbf{z}_1^{(L)}, \nabla_{\mathbf{a}^{(I)}}\mathbf{z}_2^{(L)}, \ldots, \nabla_{\mathbf{a}^{(I)}}\mathbf{z}_{N_L}^{(L)}\right]^T, \qquad I \in [L-1] \tag{35}$$

$\nabla_{\mathbf{x}}^2 \mathbf{z}_i^{(L)}$ can be written in terms of the activations of the previous layer using the following formula:

$$\nabla_{\mathbf{x}}^2 \mathbf{z}_i^{(L)} = \sum_{j=1}^{N_{I-1}} \mathbf{W}_{i,j}^{(L)}\left(\nabla_{\mathbf{x}}^2 \mathbf{a}_j^{(L-1)}\right) \tag{36}$$

Using the chain rule of the Hessian and $\mathbf{a}^{(I)} = \sigma(\mathbf{z}^{(I)})$, we can write $\nabla_{\mathbf{x}}^2 \mathbf{a}_j^{(L-1)}$ in terms of $\nabla_{\mathbf{x}}\mathbf{z}_j^{(L-1)}$ and $\nabla_{\mathbf{x}}^2 \mathbf{z}_j^{(L-1)}$ as the following:

$$\nabla_{\mathbf{x}}^2 \mathbf{a}_j^{(L-1)} = \sigma''\left(\mathbf{z}_j^{(L-1)}\right)\left(\nabla_{\mathbf{x}}\mathbf{z}_j^{(L-1)}\right)\left(\nabla_{\mathbf{x}}\mathbf{z}_j^{(L-1)}\right)^T + \sigma'\left(\mathbf{z}_j^{(L-1)}\right)\left(\nabla_{\mathbf{x}}^2 \mathbf{z}_j^{(L-1)}\right) \tag{37}$$

Replacing $\nabla_{\mathbf{x}}^2 \mathbf{a}_j^{(L-1)}$ using equation (37) into equation (36), we get:

$$\nabla_{\mathbf{x}}^2 \mathbf{z}_i^{(L)} = \sum_{j=1}^{N_{L-1}} \mathbf{W}_{i,j}^{(L)}\left[\sigma''\left(\mathbf{z}_j^{(L-1)}\right)\left(\nabla_{\mathbf{x}}\mathbf{z}_j^{(L-1)}\right)\left(\nabla_{\mathbf{x}}\mathbf{z}_j^{(L-1)}\right)^T + \sigma'\left(\mathbf{z}_j^{(L-1)}\right)\left(\nabla_{\mathbf{x}}^2 \mathbf{z}_j^{(L-1)}\right)\right]$$

$$\nabla_{\mathbf{x}}^2 \mathbf{z}_i^{(L)} = \sum_{j=1}^{N_{L-1}} \mathbf{W}_{i,j}^{(L)}\sigma''\left(\mathbf{z}_j^{(L-1)}\right)\left(\nabla_{\mathbf{x}}\mathbf{z}_j^{(L-1)}\right)\left(\nabla_{\mathbf{x}}\mathbf{z}_j^{(L-1)}\right)^T$$

$$+ \sum_{j=1}^{N_{L-1}} \mathbf{W}_{i,j}^{(L)}\sigma'\left(\mathbf{z}_j^{(L-1)}\right)\left(\nabla_{\mathbf{x}}^2 \mathbf{z}_j^{(L-1)}\right) \tag{38}$$

For each $I \in [2, L]$, $i \in N_I$, we define the matrix $\mathbf{A}_i^{(I)}$ as the following:

$$\nabla_{\mathbf{x}}^2 \left(\mathbf{z}_i^{(I)}\right) = \underbrace{\sum_{j=1}^{N_{I-1}} \mathbf{W}_{i,j}^{(I)}\sigma''\left(\mathbf{z}_j^{(I-1)}\right)\left(\nabla_{\mathbf{x}}\mathbf{z}_j^{(I-1)}\right)\left(\nabla_{\mathbf{x}}\mathbf{z}_j^{(I-1)}\right)^T}_{\mathbf{A}_i^{(I)}} + \sum_{j=1}^{N_{I-1}} \mathbf{W}_{i,j}^{(I)}\sigma'\left(\mathbf{z}_j^{(I-1)}\right)\left(\nabla_{\mathbf{x}}^2 \mathbf{z}_j^{(I-1)}\right)$$

$$\tag{39}$$

$$\mathbf{A}_i^{(I)} = \sum_{j=1}^{N_{I-1}} \mathbf{W}_{i,j}^{(I)}\sigma''\left(\mathbf{z}_j^{(I-1)}\right)\left(\nabla_{\mathbf{x}}\mathbf{z}_j^{(I-1)}\right)\left(\nabla_{\mathbf{x}}\mathbf{z}_j^{(I-1)}\right)^T \tag{40}$$

Substituting $\mathbf{A}_i^{(L)}$ using equation (40) into equation (38), we get:

$$\nabla_{\mathbf{x}}^2 \left(\mathbf{z}_i^{(L)}\right) = \mathbf{A}_i^{(L)} + \sum_{j=1}^{N_{I-1}} \mathbf{W}_{i,j}^{(I)}\sigma'\left(\mathbf{z}_j^{(I-1)}\right)\left(\nabla_{\mathbf{x}}^2 \mathbf{z}_j^{(I-1)}\right) \tag{41}$$

We first simplify the expression for $\mathbf{A}_i^{(L)}$. Note that $\mathbf{A}_i^{(L)}$ is a sum of symmetric rank one matrices $\left(\nabla_{\mathbf{x}}\mathbf{z}_j^{(L-1)}\right)\left(\nabla_{\mathbf{x}}\mathbf{z}_j^{(L-1)}\right)^T$ with the coefficient $\mathbf{W}_{i,j}^{(L)}\sigma''\left(\mathbf{z}_j^{(L-1)}\right)$ for each $j$. We create a diagonal

matrix for the coefficients and another matrix $\mathbf{B}^{(L-1)}$ such that each $j^{th}$ row of $\mathbf{B}^{(L-1)}$ is the vector $\nabla_{\mathbf{x}}\mathbf{z}_j^{(L-1)}$. This leads to the following equation:

$$\mathbf{A}_i^{(L)} = \sum_{j=1}^{N_{L-1}} \mathbf{W}_{i,j}^{(L)} \sigma'' \left(\mathbf{z}_j^{(L-1)}\right) \left(\nabla_{\mathbf{x}}\mathbf{z}_j^{(L-1)}\right) \left(\nabla_{\mathbf{x}}\mathbf{z}_j^{(L-1)}\right)^T$$

$$= \left(\mathbf{B}^{(L-1)}\right)^T diag\left(\mathbf{W}_i^{(L)} \odot \sigma''\left(\mathbf{z}^{(L-1)}\right)\right) \mathbf{B}^{(L-1)} \tag{42}$$

$\mathbf{B}^{(I)}$ where $I \in [L]$ is a matrix of size $N_I \times D$ defined as follows:

$$\mathbf{B}^{(I)} = \left[\nabla_{\mathbf{x}}\mathbf{z}_1^{(I)}, \nabla_{\mathbf{x}}\mathbf{z}_2^{(I)}, \dots, \nabla_{\mathbf{x}}\mathbf{z}_{N_I}^{(I)}\right]^T, \qquad I \in [L]$$

Thus $\mathbf{B}^{(I)}$ is the jacobian of $\mathbf{z}^{(I)}$ with respect to the input $\mathbf{x}$.
Using the chain rule of the gradient, we have the following properties of $\mathbf{B}^{(I)}$:

$$\mathbf{B}^{(1)} = \mathbf{W}^{(1)} \tag{43}$$

$$\mathbf{B}^{(I)} = \mathbf{W}^{(I)} diag\left(\sigma'\left(\mathbf{z}^{(I-1)}\right)\right) \mathbf{B}^{(I-1)} \tag{44}$$

Similarly, $\mathbf{F}^{(I,J)}$ where $I \in [L]$, $J \in [I-1]$ is a matrix of size $N_I \times N_J$ defined as follows:

$$\mathbf{F}^{(I,J)} = \left[\nabla_{\mathbf{a}^{(J)}}\mathbf{z}_1^{(I)}, \nabla_{\mathbf{a}^{(J)}}\mathbf{z}_2^{(I)}, \dots, \nabla_{\mathbf{a}^{(J)}}\mathbf{z}_{N_I}^{(I)}\right]^T, \qquad I \in [L], J \in [I-1]$$

Thus $\mathbf{F}^{(I,J)}$ is the jacobian of $\mathbf{z}^{(I)}$ with respect to the activations $\mathbf{a}^{(J)}$.
Using the chain rule of the gradient, we have the following properties for $\mathbf{F}^{(L,I)}$:

$$\mathbf{F}^{(L,L-1)} = \mathbf{W}^{(L)} \tag{45}$$

$$\mathbf{F}^{(L,I)} = \mathbf{W}^{(L)} diag\left(\sigma'\left(\mathbf{z}^{(L-1)}\right)\right) \mathbf{F}^{(L-1,I)} \tag{46}$$

Recall that in our notation: For a matrix $\mathbf{E}$, $\mathbf{E}_i$ denotes the column vector constructed by taking the transpose of the $i^{th}$ row of the matrix $\mathbf{E}$. Thus $i^{th}$ row of $\mathbf{W}^{(L)}$ is $\left(\mathbf{W}_i^{(L)}\right)^T$ and $\mathbf{F}^{(L,I)}$ is $\left(\mathbf{F}_i^{(L,I)}\right)^T$. Equating the $i^{th}$ rows in equation (46), we get:

$$\left(\mathbf{F}_i^{(L,I)}\right)^T = \left(\mathbf{W}_i^{(L)}\right)^T diag\left(\sigma'\left(\mathbf{z}^{(L-1)}\right)\right) \mathbf{F}^{(L-1,I)}$$

Taking the transpose of both the sides and expressing the RHS as a summation, we get:

$$\mathbf{F}_i^{(L,I)} = \left(\left(\mathbf{W}_i^{(L)}\right)^T diag\left(\sigma'\left(\mathbf{z}^{(L-1)}\right)\right) \mathbf{F}^{(L-1,I)}\right)^T = \sum_{j=1}^{N_{L-1}} \mathbf{W}_{i,j}^{(L)} \sigma'\left(\mathbf{z}_j^{(L-1)}\right) \mathbf{F}_j^{(L-1,I)} \tag{47}$$

Substituting $\mathbf{W}^{(L)}$ using equation (45) into equation (42), we get:

$$\mathbf{A}_i^{(L)} = \left(\mathbf{B}^{(L-1)}\right)^T diag\left(\mathbf{F}_i^{(L,L-1)} \odot \sigma''\left(\mathbf{z}^{(L-1)}\right)\right) \mathbf{B}^{(L-1)} \tag{48}$$

Substituting $\mathbf{A}_i^{(L)}$ using equation (48) into (41), we get:

$$\nabla_{\mathbf{x}}^2\mathbf{z}_i^{(L)} = \left(\mathbf{B}^{(L-1)}\right)^T diag\left(\mathbf{F}_i^{(L,L-1)} \odot \sigma''\left(\mathbf{z}^{(L-1)}\right)\right) \mathbf{B}^{(L-1)}$$

$$+ \sum_{j=1}^{N_{L-1}} \mathbf{W}_{i,j}^{(L)} \sigma'\left(\mathbf{z}_j^{(L-1)}\right) \left(\nabla_{\mathbf{x}}^2\mathbf{z}_j^{(L-1)}\right) \tag{49}$$

Thus, equation (49) allows us to write the hessian of $i^{th}$ unit at layer $L$, i.e $\left(\nabla_{\mathbf{x}}^2\mathbf{z}_i^{(L)}\right)$ in terms of the hessian of $j^{th}$ unit at layer $L-1$, i.e $\left(\nabla_{\mathbf{x}}^2\mathbf{z}_j^{(L-1)}\right)$.
We will prove the following using induction:

$$\nabla_{\mathbf{x}}^2\mathbf{z}_i^{(L)} = \sum_{I=1}^{L-1} \left(\mathbf{B}^{(I)}\right)^T diag\left(\mathbf{F}_i^{(L,I)} \odot \sigma''\left(\mathbf{z}^{(I)}\right)\right) \mathbf{B}^{(I)} \tag{50}$$

Note that for $L = 2$, $\nabla_{\mathbf{x}}^2 \mathbf{z}_j^{(L-1)} = 0$, $\forall j \in N_1$. Thus using (49) we have:

$$\nabla_{\mathbf{x}}^2 \mathbf{z}_i^{(2)} = \left(\mathbf{B}^{(1)}\right)^T diag\left(\mathbf{F}_i^{(2,1)} \odot \sigma''\left(\mathbf{z}^{(1)}\right)\right) \mathbf{B}^{(1)}$$

Hence the induction hypothesis (50) is true for $L = 2$.
Now we will assume (50) is true for $L - 1$. Thus we have:

$$\nabla_{\mathbf{x}}^2 \mathbf{z}_j^{(L-1)} = \sum_{I=1}^{L-2} \left(\mathbf{B}^{(I)}\right)^T diag\left(\mathbf{F}_j^{(L-1,I)} \odot \sigma''\left(\mathbf{z}^{(I)}\right)\right) \mathbf{B}^{(I)} \qquad \forall j \in N_{L-1} \qquad (51)$$

We will prove the same for $L$.
Using equation (49), we have:

$$\nabla_{\mathbf{x}}^2 \mathbf{z}_i^{(L)} = \left(\mathbf{B}^{(L-1)}\right)^T diag\left(\mathbf{F}_i^{(L,L-1)} \odot \sigma''\left(\mathbf{z}^{(L-1)}\right)\right) \mathbf{B}^{(L-1)}$$
$$+ \sum_{j=1}^{N_{L-1}} \mathbf{W}_{i,j}^{(L)} \sigma'\left(\mathbf{z}_j^{(L-1)}\right)\left(\nabla_{\mathbf{x}}^2 \mathbf{z}_j^{(L-1)}\right)$$

In the next set of steps, we will be working with the second term of the above equation, i.e $\sum_{j=1}^{N_{L-1}} \mathbf{W}_{i,j}^{(L)} \sigma'(\mathbf{z}_j^{(L-1)})(\nabla_{\mathbf{x}}^2 \mathbf{z}_j^{(L-1)})$:

Substituting $\nabla_{\mathbf{x}}^2 \mathbf{z}_j^{(L-1)}$ using equation (51) we get:

$$\nabla_{\mathbf{x}}^2 \mathbf{z}_i^{(L)} = \left(\mathbf{B}^{(L-1)}\right)^T diag\left(\mathbf{F}_i^{(L,L-1)} \odot \sigma''\left(\mathbf{z}^{(L-1)}\right)\right) \mathbf{B}^{(L-1)}$$
$$+ \sum_{j=1}^{N_{L-1}} \mathbf{W}_{i,j}^{(L)} \sigma'\left(\mathbf{z}_j^{(L-1)}\right)\left(\sum_{I=1}^{L-2} \left(\mathbf{B}^{(I)}\right) diag\left(\mathbf{F}_j^{(L-1,I)} \odot \sigma''\left(\mathbf{z}^{(I)}\right)\right)\left(\mathbf{B}^{(I)}\right)^T\right)$$

Combining the two summations in the second term, we get:

$$\nabla_{\mathbf{x}}^2 \mathbf{z}_i^{(L)} = \left(\mathbf{B}^{(L-1)}\right)^T diag\left(\mathbf{F}_i^{(L,L-1)} \odot \sigma''\left(\mathbf{z}^{(L-1)}\right)\right) \mathbf{B}^{(L-1)}$$
$$+ \sum_{j=1}^{N_{L-1}} \sum_{I=1}^{L-2} \mathbf{W}_{i,j}^{(L)} \sigma'\left(\mathbf{z}_j^{(L-1)}\right)\left(\mathbf{B}^{(I)}\right)^T diag\left(\mathbf{F}_j^{(L-1,I)} \odot \sigma''\left(\mathbf{z}^{(I)}\right)\right) \mathbf{B}^{(I)}$$

Exchanging the summation over $I$ and summation over $j$:

$$\nabla_{\mathbf{x}}^2 \mathbf{z}_i^{(L)} = \left(\mathbf{B}^{(L-1)}\right)^T diag\left(\mathbf{F}_i^{(L,L-1)} \odot \sigma''\left(\mathbf{z}^{(L-1)}\right)\right) \mathbf{B}^{(L-1)}$$
$$+ \sum_{I=1}^{L-2} \sum_{j=1}^{N_{L-1}} \mathbf{W}_{i,j}^{(L)} \sigma'\left(\mathbf{z}_j^{(L-1)}\right)\left(\mathbf{B}^{(I)}\right)^T diag\left(\mathbf{F}_j^{(L-1,I)} \odot \sigma''\left(\mathbf{z}^{(I)}\right)\right) \mathbf{B}^{(I)}$$

Since $\mathbf{B}^{(I)}$ is independent of $j$, we take it out of the summation over $j$:

$$\nabla_{\mathbf{x}}^2 \mathbf{z}_i^{(L)} = \left(\mathbf{B}^{(L-1)}\right)^T diag\left(\mathbf{F}_i^{(L,L-1)} \odot \sigma''\left(\mathbf{z}^{(L-1)}\right)\right) \mathbf{B}^{(L-1)}$$
$$+ \sum_{I=1}^{L-2} \left(\mathbf{B}^{(I)}\right)^T \left(\sum_{j=1}^{N_{L-1}} \mathbf{W}_{i,j}^{(L)} \sigma'\left(\mathbf{z}_j^{(L-1)}\right) diag\left(\mathbf{F}_j^{(L-1,I)} \odot \sigma''\left(\mathbf{z}^{(I)}\right)\right)\right) \mathbf{B}^{(I)}$$

Using the property, $\alpha\left(diag(\mathbf{u})\right) + \beta\left(diag(\mathbf{v})\right) = diag\left(\alpha\mathbf{u} + \beta\mathbf{v}\right)$ $\forall \alpha, \beta \in \mathbb{R}, \mathbf{u}, \mathbf{v} \in \mathbb{R}^n$; we can move the summation inside the diagonal:

$$\nabla_{\mathbf{x}}^2 \mathbf{z}_i^{(L)} = \left(\mathbf{B}^{(L-1)}\right)^T diag\left(\mathbf{F}_i^{(L,L-1)} \odot \sigma''\left(\mathbf{z}^{(L-1)}\right)\right) \mathbf{B}^{(L-1)}$$
$$+ \sum_{I=1}^{L-2} \left(\mathbf{B}^{(I)}\right)^T diag\left[\sum_{j=1}^{N_{L-1}} \mathbf{W}_{i,j}^{(L)} \sigma'\left(\mathbf{z}_j^{(L-1)}\right)\left(\mathbf{F}_j^{(L-1,I)} \odot \sigma''\left(\mathbf{z}^{(I)}\right)\right)\right] \mathbf{B}^{(I)}$$

Since $\sigma''\left(\mathbf{z}^{(I)}\right)$ is independent of $j$, we can take it out of the summation over $j$:

$$\nabla_{\mathbf{x}}^2 \mathbf{z}_i^{(L)} = \left(\mathbf{B}^{(L-1)}\right)^T diag\left(\mathbf{F}_i^{(L,L-1)} \odot \sigma''\left(\mathbf{z}^{(L-1)}\right)\right) \mathbf{B}^{(L-1)}$$
$$+ \sum_{I=1}^{L-2} \left(\mathbf{B}^{(I)}\right)^T diag\left[\left(\sum_{j=1}^{N_{L-1}} \mathbf{W}_{i,j}^{(L)} \sigma'\left(\mathbf{z}_j^{(L-1)}\right) \mathbf{F}_j^{(L-1,I)}\right) \odot \sigma''\left(\mathbf{z}^{(I)}\right)\right] \mathbf{B}^{(I)}$$

Using equation (47), we can replace $\sum_{j=1}^{N_{L-1}} \mathbf{W}_{i,j}^{(L)} \sigma' \left( \mathbf{z}_j^{(L-1)} \right) \mathbf{F}_j^{(L-1,I)}$ with $\mathbf{F}_i^{(L,I)}$:

$$\nabla_\mathbf{x}^2 \mathbf{z}_i^{(L)} = \left( \mathbf{B}^{(L-1)} \right)^T diag \left( \mathbf{F}_i^{(L,L-1)} \odot \sigma'' \left( \mathbf{z}^{(L-1)} \right) \right) \mathbf{B}^{(L-1)}$$

$$+ \sum_{I=1}^{L-2} \left( \mathbf{B}^{(I)} \right)^T diag \left( \mathbf{F}_i^{(L,I)} \odot \sigma'' \left( \mathbf{z}^{(I)} \right) \right) \mathbf{B}^{(I)}$$

$$\nabla_\mathbf{x}^2 \mathbf{z}_i^{(L)} = \sum_{I=1}^{L-1} \left( \mathbf{B}^{(I)} \right)^T diag \left( \mathbf{F}_i^{(L,I)} \odot \sigma'' \left( \mathbf{z}^{(I)} \right) \right) \mathbf{B}^{(I)}$$

### D.4 Proof of Theorem 3

Using Lemma 1, we have the following formula for $\nabla_\mathbf{x}^2 \left( \mathbf{z}_y^{(2)} - \mathbf{z}_t^{(2)} \right)$:

$$\nabla_\mathbf{x}^2 \left( \mathbf{z}_y^{(2)} - \mathbf{z}_t^{(2)} \right) = \left( \mathbf{W}^{(1)} \right)^T diag \left( \left( \mathbf{W}_y^{(2)} - \mathbf{W}_t^{(2)} \right) \odot \sigma'' \left( \mathbf{z}^{(1)} \right) \right) \mathbf{W}^{(1)}$$

$$= \sum_{i=1}^{N_1} \left( \mathbf{W}_{y,i}^{(2)} - \mathbf{W}_{t,i}^{(2)} \right) \sigma'' \left( \mathbf{z}_i^{(1)} \right) \mathbf{W}_i^{(1)} (\mathbf{W}_i^{(1)})^T \tag{52}$$

We are also given that the activation function $\sigma$ satisfies the following property:

$$h_L \le \sigma'' (x) \le h_U \quad \forall x \in \mathbb{R} \tag{53}$$

(a) We have to prove the following linear matrix inequalities (LMIs):

$$\mathbf{N} \preccurlyeq \nabla_\mathbf{x}^2 \left( \mathbf{z}_y^{(2)} - \mathbf{z}_t^{(2)} \right) \preccurlyeq \mathbf{P} \quad \forall \mathbf{x} \in \mathbb{R}^D \tag{54}$$

where $\mathbf{P}$ and $\mathbf{N}$ are given as following:

$$\mathbf{P} = \sum_{i=1}^{N_1} p_i \left( \mathbf{W}_{y,i}^{(2)} - \mathbf{W}_{t,i}^{(2)} \right) \mathbf{W}_i^{(1)} \left( \mathbf{W}_i^{(1)} \right)^T \tag{55}$$

$$\mathbf{N} = \sum_{i=1}^{N_1} n_i \left( \mathbf{W}_{y,i}^{(2)} - \mathbf{W}_{t,i}^{(2)} \right) \mathbf{W}_i^{(1)} \left( \mathbf{W}_i^{(1)} \right)^T \tag{56}$$

$$p_i = \begin{cases} h_U, & \mathbf{W}_{y,i}^{(2)} - \mathbf{W}_{t,i}^{(2)} \ge 0 \\ h_L, & \mathbf{W}_{y,i}^{(2)} - \mathbf{W}_{t,i}^{(2)} \le 0 \end{cases}, \quad n_i = \begin{cases} h_L, & \mathbf{W}_{y,i}^{(2)} - \mathbf{W}_{t,i}^{(2)} \ge 0 \\ h_U, & \mathbf{W}_{y,i}^{(2)} - \mathbf{W}_{t,i}^{(2)} \le 0 \end{cases} \tag{57}$$

We first prove: $\mathbf{N} \preccurlyeq \nabla_\mathbf{x}^2 \left( \mathbf{z}_y^{(2)} - \mathbf{z}_t^{(2)} \right) \quad \forall \mathbf{x} \in \mathbb{R}^D$:

We substitute $\nabla_\mathbf{x}^2 \left( \mathbf{z}_y^{(2)} - \mathbf{z}_t^{(2)} \right)$ and $\mathbf{N}$ from equations (52) and (56) respectively in $\nabla_\mathbf{x}^2 \left( \mathbf{z}_y^{(2)} - \mathbf{z}_t^{(2)} \right) - \mathbf{N}$:

$$\nabla_\mathbf{x}^2 \left( \mathbf{z}_y^{(2)} - \mathbf{z}_t^{(2)} \right) - \mathbf{N}$$

$$= \sum_{i=1}^{N_1} \left( \left( \mathbf{W}_{y,i}^{(2)} - \mathbf{W}_{t,i}^{(2)} \right) \sigma'' \left( \mathbf{z}_i^{(1)} \right) - \left( \mathbf{W}_{y,i}^{(2)} - \mathbf{W}_{t,i}^{(2)} \right) n_i \right) \mathbf{W}_i^{(1)} \left( \mathbf{W}_i^{(1)} \right)^T$$

$$= \sum_{i=1}^{N_1} \left( \mathbf{W}_{y,i}^{(2)} - \mathbf{W}_{t,i}^{(2)} \right) \left( \sigma'' \left( \mathbf{z}_i^{(1)} \right) - n_i \right) \mathbf{W}_i^{(1)} \left( \mathbf{W}_i^{(1)} \right)^T$$

Thus $\nabla_\mathbf{x}^2 \left( \mathbf{z}_y^{(2)} - \mathbf{z}_t^{(2)} \right) - \mathbf{N}$ is a weighted sum of symmetric rank one matrices i.e, $\mathbf{W}_i^{(1)} \left( \mathbf{W}_i^{(1)} \right)^T$ and it is PSD if and only if coefficient of each rank one matrix i.e, $\left( \mathbf{W}_{y,i}^{(2)} - \mathbf{W}_{t,i}^{(2)} \right) \left( \sigma'' \left( \mathbf{z}_i^{(1)} \right) - n_i \right)$ is positive. Using equations (53) and (57), we have

the following:

$$\left(\mathbf{W}_{y,i}^{(2)} - \mathbf{W}_{t,i}^{(2)}\right) \geq 0 \implies n_i = h_L \implies \left(\sigma''\left(\mathbf{z}_i^{(1)}\right) - n_i\right) \geq 0 \qquad \forall i \in [N_1],\ \forall \mathbf{x} \in \mathbb{R}^D$$

$$\left(\mathbf{W}_{y,i}^{(2)} - \mathbf{W}_{t,i}^{(2)}\right) \leq 0 \implies n_i = h_U \implies \left(\sigma''\left(\mathbf{z}_i^{(1)}\right) - n_i\right) \leq 0 \qquad \forall i \in [N_1],\ \forall \mathbf{x} \in \mathbb{R}^D$$

$$\implies \left(\mathbf{W}_{y,i}^{(2)} - \mathbf{W}_{t,i}^{(2)}\right)\left(\sigma''\left(\mathbf{z}_i^{(1)}\right) - n_i\right) \geq 0 \qquad \forall i \in [N_1],\ \forall \mathbf{x} \in \mathbb{R}^D \tag{58}$$

Thus $\nabla_{\mathbf{x}}^2\left(\mathbf{z}_y^{(2)} - \mathbf{z}_t^{(2)}\right) - \mathbf{N}$ is a PSD matrix i.e:

$$\nabla_{\mathbf{x}}^2\left(\mathbf{z}_y^{(2)} - \mathbf{z}_t^{(2)}\right) - \mathbf{N} = \sum_{i=1}^{N_1} \underbrace{\left(\mathbf{W}_{y,i}^{(2)} - \mathbf{W}_{t,i}^{(2)}\right)\left(\sigma''\left(\mathbf{z}_i^{(1)}\right) - n_i\right)}_{\text{always positive using eq. (58)}} \mathbf{W}_i^{(1)}\left(\mathbf{W}_i^{(1)}\right)^T \succcurlyeq 0 \qquad \forall \mathbf{x} \in \mathbb{R}^D$$

$$\implies \mathbf{N} \preccurlyeq \nabla_{\mathbf{x}}^2\left(\mathbf{z}_y^{(2)} - \mathbf{z}_t^{(2)}\right) \qquad \forall \mathbf{x} \in \mathbb{R}^D \tag{59}$$

Now we prove that $\nabla_{\mathbf{x}}^2\left(\mathbf{z}_y^{(2)} - \mathbf{z}_t^{(2)}\right) \preccurlyeq \mathbf{P} \quad \forall \mathbf{x} \in \mathbb{R}^D$:

We substitute $\nabla_{\mathbf{x}}^2\left(\mathbf{z}_y^{(2)} - \mathbf{z}_t^{(2)}\right)$ and $\mathbf{P}$ from equations (52) and (56) respectively in $\mathbf{P} - \nabla_{\mathbf{x}}^2\left(\mathbf{z}_y^{(2)} - \mathbf{z}_t^{(2)}\right)$:

$$\mathbf{P} - \nabla_{\mathbf{x}}^2\left(\mathbf{z}_y^{(2)} - \mathbf{z}_t^{(2)}\right) = \sum_{i=1}^{N_1}\left(\left(\mathbf{W}_{y,i}^{(2)} - \mathbf{W}_{t,i}^{(2)}\right)p_i - \left(\mathbf{W}_{y,i}^{(2)} - \mathbf{W}_{t,i}^{(2)}\right)\sigma''\left(\mathbf{z}_i^{(1)}\right)\right)\mathbf{W}_i^{(1)}(\mathbf{W}_i^{(1)})^T$$

$$= \sum_{i=1}^{N_1}\left(\mathbf{W}_{y,i}^{(2)} - \mathbf{W}_{t,i}^{(2)}\right)\left(p_i - \sigma''\left(\mathbf{z}_i^{(1)}\right)\right)\mathbf{W}_i^{(1)}(\mathbf{W}_i^{(1)})^T$$

Thus $\mathbf{P} - \nabla_{\mathbf{x}}^2\left(\mathbf{z}_y^{(2)} - \mathbf{z}_t^{(2)}\right)$ is a weighted sum of symmetric rank one matrices i.e, $\mathbf{W}_i^{(1)}\left(\mathbf{W}_i^{(1)}\right)^T$ and it is PSD if and only if coefficient of each rank one matrix i.e, $\left(\mathbf{W}_{y,i}^{(2)} - \mathbf{W}_{t,i}^{(2)}\right)\left(p_i - \sigma''\left(\mathbf{z}_i^{(1)}\right)\right)$ is positive. Using equations (53) and (57), we have the following:

$$\left(\mathbf{W}_{y,i}^{(2)} - \mathbf{W}_{t,i}^{(2)}\right) \geq 0 \implies p_i = h_U \implies \left(p_i - \sigma''\left(\mathbf{z}_i^{(1)}\right)\right) \geq 0 \qquad \forall i \in N_1,\ \mathbf{x} \in \mathbb{R}^D$$

$$\left(\mathbf{W}_{y,i}^{(2)} - \mathbf{W}_{t,i}^{(2)}\right) \leq 0 \implies p_i = h_L \implies \left(p_i - \sigma''\left(\mathbf{z}_i^{(1)}\right)\right) \leq 0 \qquad \forall i \in N_1,\ \mathbf{x} \in \mathbb{R}^D$$

$$\implies \left(\mathbf{W}_{y,i}^{(2)} - \mathbf{W}_{t,i}^{(2)}\right)\left(p_i - \sigma''\left(\mathbf{z}_i^{(1)}\right)\right) \geq 0 \qquad \forall i \in [N_1],\ \mathbf{x} \in \mathbb{R}^D \tag{60}$$

Thus $\mathbf{P} - \nabla_{\mathbf{x}}^2\left(\mathbf{z}_y^{(2)} - \mathbf{z}_t^{(2)}\right)$ is PSD matrix i.e:

$$\mathbf{P} - \nabla_{\mathbf{x}}^2\left(\mathbf{z}_y^{(2)} - \mathbf{z}_t^{(2)}\right)$$

$$= \sum_{i=1}^{N_1} \underbrace{\left(\mathbf{W}_{y,i}^{(2)} - \mathbf{W}_{t,i}^{(2)}\right)\left(p_i - \sigma''\left(\mathbf{z}_i^{(1)}\right)\right)}_{\text{always positive using eq. (60)}} \mathbf{W}_i^{(1)}\left(\mathbf{W}_i^{(1)}\right)^T \succcurlyeq 0 \qquad \forall \mathbf{x} \in \mathbb{R}^D$$

$$\implies \mathbf{P} \succcurlyeq \nabla_{\mathbf{x}}^2\left(\mathbf{z}_y^{(2)} - \mathbf{z}_t^{(2)}\right) \qquad \forall \mathbf{x} \in \mathbb{R}^D \tag{61}$$

Thus by proving the LMIs (59) and (61), we prove (54).

(b) We have to prove that if $h_U \geq 0$ and $h_L \leq 0$, $\mathbf{P}$ is a PSD matrix, $\mathbf{N}$ is a NSD matrix. We are given $h_U \geq 0$, $h_L \leq 0$. Using equation (57), we have the following:

$$\left(\mathbf{W}_{y,i}^{(2)} - \mathbf{W}_{t,i}^{(2)}\right) \geq 0 \implies p_i = h_U \geq 0 \implies p_i\left(\mathbf{W}_{y,i}^{(2)} - \mathbf{W}_{t,i}^{(2)}\right) \geq 0$$

$$\left(\mathbf{W}_{y,i}^{(2)} - \mathbf{W}_{t,i}^{(2)}\right) \leq 0 \implies p_i = h_L \leq 0 \implies p_i\left(\mathbf{W}_{y,i}^{(2)} - \mathbf{W}_{t,i}^{(2)}\right) \geq 0$$

$$\implies p_i\left(\mathbf{W}_{y,i}^{(2)} - \mathbf{W}_{t,i}^{(2)}\right) \geq 0 \qquad \forall i \in [N_1] \tag{62}$$

Thus $\mathbf{P}$ is a weighted sum of symmetric rank one matrices i.e, $\mathbf{W}_i^{(1)}\left(\mathbf{W}_i^{(1)}\right)^T$ and each coefficient $p_i\left(\mathbf{W}_{y,i}^{(2)} - \mathbf{W}_{t,i}^{(2)}\right)$ is positive.

$$\mathbf{P} = \sum_{i=1}^{N_1} \underbrace{p_i\left(\mathbf{W}_{y,i}^{(2)} - \mathbf{W}_{t,i}^{(2)}\right)}_{\text{always positive using eq. (62)}} \mathbf{W}_i^{(1)}\left(\mathbf{W}_i^{(1)}\right)^T \succcurlyeq 0$$

Using equation (57), we have the following:

$$\left(\mathbf{W}_{y,i}^{(2)} - \mathbf{W}_{t,i}^{(2)}\right) \geq 0 \implies n_i = h_L \leq 0 \implies n_i\left(\mathbf{W}_{y,i}^{(2)} - \mathbf{W}_{t,i}^{(2)}\right) \leq 0$$

$$\left(\mathbf{W}_{y,i}^{(2)} - \mathbf{W}_{t,i}^{(2)}\right) \leq 0 \implies n_i = h_U \geq 0 \implies n_i\left(\mathbf{W}_{y,i}^{(2)} - \mathbf{W}_{t,i}^{(2)}\right) \leq 0$$

$$\implies n_i\left(\mathbf{W}_{y,i}^{(2)} - \mathbf{W}_{t,i}^{(2)}\right) \geq 0 \qquad \forall i \in [N_1] \tag{63}$$

$$\mathbf{N} = \sum_{i=1}^{N_1} \underbrace{n_i\left(\mathbf{W}_{y,i}^{(2)} - \mathbf{W}_{t,i}^{(2)}\right)}_{\text{always positive using eq. (63)}} \mathbf{W}_i^{(1)}\left(\mathbf{W}_i^{(1)}\right)^T \preccurlyeq 0$$

Thus $\mathbf{P}$ is a PSD and $\mathbf{N}$ is a NSD matrix if $h_U \geq 0$ and $h_L \leq 0$.

(c) We have to prove the following global bounds on the eigenvalues of $\nabla_{\mathbf{x}}^2(\mathbf{z}_y^{(2)} - \mathbf{z}_t^{(2)})$:

$$m\mathbf{I} \preccurlyeq \nabla_{\mathbf{x}}^2\left(\mathbf{z}_y^{(2)} - \mathbf{z}_t^{(2)}\right) \preccurlyeq M\mathbf{I}, \qquad \text{where } M = \max_{\|\mathbf{v}\|=1} \mathbf{v}^T\mathbf{P}\mathbf{v}, \; m = \min_{\|\mathbf{v}\|=1} \mathbf{v}^T\mathbf{N}\mathbf{v}$$

Since $\nabla_{\mathbf{x}}^2\left(\mathbf{z}_y^{(2)} - \mathbf{z}_t^{(2)}\right) \preccurlyeq \mathbf{P} \quad \forall \mathbf{x} \in \mathbb{R}^D$:

$$\mathbf{v}^T\left[\nabla_{\mathbf{x}}^2\left(\mathbf{z}_y^{(2)} - \mathbf{z}_t^{(2)}\right)\right]\mathbf{v} \leq \mathbf{v}^T\mathbf{P}\mathbf{v} \qquad \forall \mathbf{v} \in \mathbb{R}^D, \; \forall \mathbf{x} \in \mathbb{R}^D \tag{64}$$

Let $\mathbf{v}^*$, $\mathbf{x}^*$ be vectors such that:

$$(\mathbf{v}^*)^T\left[\nabla_{\mathbf{x}^*}^2\left(\mathbf{z}_y^{(2)} - \mathbf{z}_t^{(2)}\right)\right]\mathbf{v}^* = \max_{\mathbf{x}} \max_{\|\mathbf{v}\|=1} \mathbf{v}^T\left[\nabla_{\mathbf{x}}^2\left(\mathbf{z}_y^{(2)} - \mathbf{z}_t^{(2)}\right)\right]\mathbf{v}$$

Thus using inequality (64):

$$(\mathbf{v}^*)^T\left[\nabla_{\mathbf{x}^*}^2\left(\mathbf{z}_y^{(2)} - \mathbf{z}_t^{(2)}\right)\right]\mathbf{v}^* \leq (\mathbf{v}^*)^T\mathbf{P}\mathbf{v}^* \leq \max_{\|\mathbf{v}\|=1}\mathbf{v}^T\mathbf{P}\mathbf{v}$$

$$\implies \max_{\mathbf{x}} \max_{\|\mathbf{v}\|=1} \mathbf{v}^T\left[\nabla_{\mathbf{x}}^2\left(\mathbf{z}_y^{(2)} - \mathbf{z}_t^{(2)}\right)\right]\mathbf{v} \leq \max_{\|\mathbf{v}\|=1}\mathbf{v}^T\mathbf{P}\mathbf{v} \tag{65}$$

Since $\mathbf{N} \preccurlyeq \nabla_{\mathbf{x}}^2\left(\mathbf{z}_y^{(2)} - \mathbf{z}_t^{(2)}\right) \quad \forall \mathbf{x} \in \mathbb{R}^D$:

$$\mathbf{v}^T\mathbf{N}\mathbf{v} \leq \mathbf{v}^T\left[\nabla_{\mathbf{x}}^2\left(\mathbf{z}_y^{(2)} - \mathbf{z}_t^{(2)}\right)\right]\mathbf{v} \qquad \forall \mathbf{v} \in \mathbb{R}^D, \; \forall \mathbf{x} \in \mathbb{R}^D \tag{66}$$

Let $\mathbf{v}^*$, $\mathbf{x}^*$ be vectors such that:

$$(\mathbf{v}^*)^T\left[\nabla_{\mathbf{x}^*}^2\left(\mathbf{z}_y^{(2)} - \mathbf{z}_t^{(2)}\right)\right]\mathbf{v}^* = \min_{\mathbf{x}} \min_{\|\mathbf{v}\|=1} \mathbf{v}^T\left[\nabla_{\mathbf{x}}^2\left(\mathbf{z}_y^{(2)} - \mathbf{z}_t^{(2)}\right)\right]\mathbf{v}$$

Thus using inequality (66):

$$(\mathbf{v}^*)^T\left[\nabla_{\mathbf{x}^*}^2\left(\mathbf{z}_y^{(2)} - \mathbf{z}_t^{(2)}\right)\right]\mathbf{v}^* \geq (\mathbf{v}^*)^T\mathbf{N}\mathbf{v}^* \geq \min_{\|\mathbf{v}\|=1}\mathbf{v}^T\mathbf{N}\mathbf{v}$$

$$\implies \min_{\mathbf{x}} \min_{\|\mathbf{v}\|=1} \mathbf{v}^T\left[\nabla_{\mathbf{x}}^2\left(\mathbf{z}_y^{(2)} - \mathbf{z}_t^{(2)}\right)\right]\mathbf{v} \geq \min_{\|\mathbf{v}\|=1}\mathbf{v}^T\mathbf{N}\mathbf{v} \tag{67}$$

Using the inequalities (65) and (67), we get:

$$m\mathbf{I} \preccurlyeq \nabla_{\mathbf{x}}^2\left(\mathbf{z}_y^{(2)} - \mathbf{z}_t^{(2)}\right) \preccurlyeq M\mathbf{I}, \qquad \text{where } M = \max_{\|\mathbf{v}\|=1} \mathbf{v}^T\mathbf{P}\mathbf{v}, \; m = \min_{\|\mathbf{v}\|=1} \mathbf{v}^T\mathbf{N}\mathbf{v}$$

### D.5   PROOF OF THEOREM 4

We are given that the activation function $\sigma$ is such that $\sigma'$, $\sigma''$ are bounded, i.e:

$$|\sigma'(x)| \le g, \ |\sigma''(x)| \le h \qquad \forall x \in \mathbb{R} \tag{68}$$

We have to prove the following:

$$\left\| \nabla_\mathbf{x}^2 \mathbf{z}_i^{(L)} \right\| \le h \sum_{I=1}^{L-1} \left( r^{(I)} \right)^2 \max_j \left( \mathbf{S}_{i,j}^{(I)} \right) \quad \forall \mathbf{x} \in \mathbb{R}^D$$

where $\mathbf{S}^{(L,I)}$ is a matrix of size $N_L \times N_I$ defined as follows:

$$\mathbf{S}^{(L,I)} = \begin{cases} \left| \mathbf{W}^{(L)} \right| & I = L - 1 \\ g \left| \mathbf{W}^{(L)} \right| \mathbf{S}^{(L-1,I)} & I \in [L-2] \end{cases} \tag{69}$$

and $r^{(I)}$ is a scalar defined as follows:

$$r^{(I)} = \begin{cases} \left\| \mathbf{W}^{(1)} \right\| & I = 1 \\ g \left\| \mathbf{W}^{(I)} \right\| r^{(I-1)} & I \in [2, L-1] \end{cases} \tag{70}$$

We will prove the same in 3 steps.
In step (a), we will prove:

$$\left| \mathbf{F}_{i,j}^{(L,I)} \right| \le \mathbf{S}_{i,j}^{(L,I)} \qquad \forall \mathbf{x} \in \mathbb{R}^D \tag{71}$$

In step (b), we will prove:

$$\left\| \mathbf{B}^{(I)} \right\| \le r^{(I)}, \qquad \forall \mathbf{x} \in \mathbb{R}^D \tag{72}$$

In step (c), we will use (a) and (b) to prove:

$$\left\| \nabla_\mathbf{x}^2 \mathbf{z}_i^{(L)} \right\| \le h \sum_{I=1}^{L-1} \left( r^{(I)} \right)^2 \max_j \left( \mathbf{S}_{i,j}^{(L,I)} \right) \tag{73}$$

Note that $\mathbf{B}^{(I)}$ and $\mathbf{F}^{(L,I)}$ are defined using (34) and (35) respectively.

(a) We have to prove that for $L \ge 2$, $I \in [L-1]$, $i \in N_L$, $j \in N_I$:

$$\left| \mathbf{F}_{i,j}^{(L,I)} \right| \le \mathbf{S}_{i,j}^{(L,I)} \qquad \forall \mathbf{x} \in \mathbb{R}^D$$

where $\mathbf{S}^{(L,I)}$ is a matrix of size $N_I \times N_J$ defined as follows:

$$\mathbf{S}^{(L,I)} = \begin{cases} \left| \mathbf{W}^{(L)} \right| & I = L - 1 \\ g \left| \mathbf{W}^{(L)} \right| \mathbf{S}^{(L-1,J)} & I \in [L-2] \end{cases}$$

We first prove the case when $I = L - 1$.
Using equation (45), $\mathbf{F}_{i,j}^{(L,L-1)} = \mathbf{W}_{i,j}^{(L)}$.
Since $\mathbf{S}_{i,j}^{(L,L-1)} = \left| \mathbf{W}_{i,j}^{(L)} \right|$:

$$\left| \mathbf{F}_{i,j}^{(L,L-1)} \right| = \mathbf{S}_{i,j}^{(L,L-1)}$$

Hence for $L \ge 2$, $I = L - 1$, we have equality in (71). Hence proved.
Now, we will use proof by induction.
To prove the base case $L = 2$, note that $I = L - 1 = 1$ is the only possible value for $I$. Thus, using the result for $I = L - 1$, the theorem holds for $L = 2$. This proves the base case.
Now we assume the induction hypothesis is true for depth $= L - 1$, $I \in [L-2]$. and prove for depth $= L$, $I \in [L-1]$. Since for $I = L - 1$, we have proven already, we prove for $I \le L - 2$.
Using equation (47), we have the following formula for $\mathbf{F}_i^{(L,I)}$:

$$\mathbf{F}_i^{(L,I)} = \sum_{k=1}^{N_{L-1}} \mathbf{W}_{i,k}^{(L)} \sigma' \left( \mathbf{z}_k^{(L-1)} \right) \mathbf{F}_k^{(L-1,I)}$$

Taking the $j^{th}$ element of the vectors on both sides:

$$\mathbf{F}_{i,j}^{(L,I)} = \sum_{k=1}^{N_{L-1}} \mathbf{W}_{i,k}^{(L)} \sigma' \left( \mathbf{z}_k^{(L-1)} \right) \mathbf{F}_{k,j}^{(L-1,I)} \tag{74}$$

By induction hypothesis, we know that:

$$\left| \mathbf{F}_{k,j}^{(L-1,I)} \right| \leq \mathbf{S}_{k,j}^{(L-1,I)} \tag{75}$$

Using the absolute value properties for equation (74), we have:

$$\left| \mathbf{F}_{i,j}^{(L,I)} \right| = \left| \sum_{k=1}^{N_{L-1}} \mathbf{W}_{i,k}^{(L)} \sigma' \left( \mathbf{z}_k^{(L-1)} \right) \mathbf{F}_{k,j}^{(L-1,I)} \right|$$

$$\left| \mathbf{F}_{i,j}^{(L,I)} \right| \leq \sum_{k=1}^{N_{L-1}} \left| \mathbf{W}_{i,k}^{(L)} \sigma' \left( \mathbf{z}_k^{(L-1)} \right) \mathbf{F}_{k,j}^{(L-1,I)} \right|$$

$$\left| \mathbf{F}_{i,j}^{(L,I)} \right| \leq \sum_{k=1}^{N_{L-1}} \left| \mathbf{W}_{i,k}^{(L)} \right| \left| \sigma' \left( \mathbf{z}_k^{(L-1)} \right) \right| \left| \mathbf{F}_{k,j}^{(L-1,I)} \right|$$

Using $|\sigma'(x)| \leq g \quad \forall x \in \mathbb{R}$ (inequality (68)) :

$$\left| \mathbf{F}_{i,j}^{(L,I)} \right| \leq g \sum_{k=1}^{N_{L-1}} \left| \mathbf{W}_{i,k}^{(L)} \right| \left| \mathbf{F}_{k,j}^{(L-1,I)} \right|$$

Using the induction hypothesis (inequality (75)):

$$\left| \mathbf{F}_{i,j}^{(L,I)} \right| \leq g \sum_{k=1}^{N_{L-1}} \left| \mathbf{W}_{i,k}^{(L)} \right| \left| \mathbf{S}_{k,j}^{(L-1,I)} \right|$$

Using equation (69) for definition of $\mathbf{S}_{i,j}^{(L,I)}$:

$$\left| \mathbf{F}_{i,j}^{(L,I)} \right| \leq \mathbf{S}_{i,j}^{(L,I)}$$

Hence we prove (71) for all $L \geq 2$ and $I \leq L - 1$ using induction.

(b) We have to prove that for $1 \leq I \leq M - 1$:

$$\left\| \mathbf{B}^{(I)} \right\| \leq r^{(I)}, \qquad \forall \mathbf{x} \in \mathbb{R}^D$$

where $r^{(I)}$ is a scalar given as follows:

$$r^{(I)} = \begin{cases} \left\| \mathbf{W}^{(1)} \right\| & I = 1 \\ g \left\| \mathbf{W}^{(I)} \right\| r^{(I-1)} & I \in [2, L-1] \end{cases}$$

Using equation (43), for $I = 1$ we have:

$$\left\| \mathbf{B}^{(1)} \right\| = \left\| \mathbf{W}^{(1)} \right\| = r^{(1)} \tag{76}$$

Using equation (44), for $I > 1$, we have:

$$\left\| \mathbf{B}^{(I)} \right\| = \left\| \mathbf{W}^{(I)} diag \left( \sigma' \left( \mathbf{z}^{(I-1)} \right) \right) \mathbf{B}^{(I-1)} \right\|$$

$$\left\| \mathbf{B}^{(I)} \right\| \leq \left\| \mathbf{W}^{(I)} \right\| \left\| diag \left( \sigma' \left( \mathbf{z}^{(I-1)} \right) \right) \right\| \left\| \mathbf{B}^{(I-1)} \right\|$$

Since $\left\| diag \left( \sigma' \left( \mathbf{z}^{(I-1)} \right) \right) \right\| = \max_j \left| \sigma' \left( \mathbf{z}_j^{(I-1)} \right) \right|$, using equation (68):

$$\left\| \mathbf{B}^{(I)} \right\| \leq g \left\| \mathbf{W}^{(I)} \right\| \left\| \mathbf{B}^{(I-1)} \right\| \leq g \left\| \mathbf{W}^{(I)} \right\| r^{(I-1)} \qquad I \geq 2 \tag{77}$$

Using inequalities (76) and (77), the proof follows using induction.

(c) We have to prove that:

$$\left\| \nabla_{\mathbf{x}}^2 \mathbf{z}_i^{(L)} \right\| \le h \sum_{I=1}^{L-1} \left( r^{(I)} \right)^2 \max_j \left( \mathbf{S}_{i,j}^{(I)} \right)$$

Using Lemma 1, we have the following equation for $\nabla_{\mathbf{x}}^2 \mathbf{z}_i^{(L)}$:

$$\nabla_{\mathbf{x}}^2 \mathbf{z}_i^{(L)} = \sum_{I=1}^{L-1} \left( \mathbf{B}^{(I)} \right)^T diag\left( \mathbf{F}_i^{(L,I)} \odot \sigma''\left( \mathbf{z}^{(I)} \right) \right) \mathbf{B}^{(I)}$$

Using the properties of norm we have:

$$\left\| \nabla_{\mathbf{x}}^2 \mathbf{z}_i^{(L)} \right\| = \left\| \sum_{I=1}^{L-1} \left( \mathbf{B}^{(I)} \right)^T diag\left( \mathbf{F}_i^{(L,I)} \odot \sigma''\left( \mathbf{z}^{(I)} \right) \right) \mathbf{B}^{(I)} \right\|$$

$$\le \sum_{I=1}^{L-1} \left\| diag\left( \mathbf{F}_i^{(L,I)} \odot \sigma''\left( \mathbf{z}^{(I)} \right) \right) \right\| \left\| \mathbf{B}^{(I)} \right\|^2$$

$$\le \sum_{I=1}^{L-1} \max_j \left( \left| \mathbf{F}_{i,j}^{(L,I)} \sigma''\left( \mathbf{z}_j^{(I)} \right) \right| \right) \left\| \mathbf{B}^{(I)} \right\|^2$$

In the last inequality, we use the property that norm of a diagonal matrix is the maximum absolute value of the diagonal element. Using the product property of absolute value, we get:

$$\left\| \nabla_{\mathbf{x}}^2 \mathbf{z}_i^{(L)} \right\| \le \sum_{I=1}^{L-1} \max_j \left( \left| \mathbf{F}_{i,j}^{(L,I)} \right| \left| \sigma''\left( \mathbf{z}_j^{(I)} \right) \right| \right) \left\| \mathbf{B}^{(I)} \right\|^2$$

Since $\left| \mathbf{F}_{i,j}^{(L,I)} \right|$ and $\left| \sigma''\left( \mathbf{z}_j^{(I)} \right) \right|$ are positive terms:

$$\left\| \nabla_{\mathbf{x}}^2 \mathbf{z}_i^{(L)} \right\| \le \sum_{I=1}^{L-1} \max_j \left( \left| \mathbf{F}_{i,j}^{(L,I)} \right| \right) \max_j \left( \left| \sigma''\left( \mathbf{z}_j^{(I)} \right) \right| \right) \left\| \mathbf{B}^{(I)} \right\|^2$$

Since $\left\| \sigma'' \right\|$ is bounded by $h$:

$$\left\| \nabla_{\mathbf{x}}^2 \mathbf{z}_i^{(L)} \right\| \le h \sum_{I=1}^{L-1} \max_j \left( \left| \mathbf{F}_{i,j}^{(L,I)} \right| \right) \left\| \mathbf{B}^{(I)} \right\|^2$$

Using inequality (71):

$$\left\| \nabla_{\mathbf{x}}^2 \mathbf{z}_i^{(L)} \right\| \le h \sum_{I=1}^{L-1} \max_j \left( \mathbf{S}_{i,j}^{(I)} \right) \left\| \mathbf{B}^{(I)} \right\|^2$$

Using inequality (72):

$$\left\| \nabla_{\mathbf{x}}^2 \mathbf{z}_i^{(L)} \right\| \le h \sum_{I=1}^{L-1} \left( r^{(I)} \right)^2 \max_j \left( \mathbf{S}_{i,j}^{(I)} \right) \qquad \forall \mathbf{x} \in \mathbb{R}^D$$

# E    COMPUTING $g, h, h_U$ AND $h_L$ FOR DIFFERENT ACTIVATION FUNCTIONS

## E.1    SOFTPLUS ACTIVATION

For softplus activation, we have the following. We use $S(x)$ to denote sigmoid:

$$\sigma(x) = \log(1 + \exp(x))$$
$$\sigma'(x) = S(x)$$
$$\sigma''(x) = S(x)(1 - S(x))$$

To bound $S(x)(1 - S(x))$, let $\alpha$ denote $S(x)$. We know that $0 \le \alpha \le 1$:

$$\alpha(1 - \alpha) = \frac{1}{4} - \left(\frac{1}{2} - \alpha\right)^2$$

Thus, $S(x)(1 - S(x))$ is maximum at $S(x) = 1/2$ and minimum at $S(x) = 0$ and $S(x) = 1$. The maximum value is 0.25 and minimum value is 0.

$$0 \le S(x)(1 - S(x)) \le 0.25 \implies 0 \le \sigma''(x) \le 0.25$$

Thus, $h_U = 0.25$, $h_L = 0$ (for use in Theorem 3) and $g = 1$, $h = 0.25$ (for use in Theorem 4).

### E.2 SIGMOID ACTIVATION

For sigmoid activation, we have the following. We use $S(x)$ to denote sigmoid:

$$\sigma(x) = S(x) = \frac{1}{1 + \exp(-x)}$$
$$\sigma'(x) = S(x)(1 - S(x))$$
$$\sigma''(x) = S(x)(1 - S(x))(1 - 2S(x))$$

The second derivative of sigmoid ($\sigma''(x)$) can be bounded using standard differentiation. Let $\alpha$ denote $S(x)$. We know that $0 \le \alpha \le 1$:

$$h_L \le \sigma''(x) \le h_U$$
$$h_L = \min_{0 \le \alpha \le 1} \alpha(1 - \alpha)(1 - 2\alpha)$$
$$h_U = \max_{0 \le \alpha \le 1} \alpha(1 - \alpha)(1 - 2\alpha)$$

To solve for both $h_L$ and $h_U$, we first differentiate $\alpha(1 - \alpha)(1 - 2\alpha)$ with respect to $\alpha$:

$$\nabla_\alpha \left(\alpha(1 - \alpha)(1 - 2\alpha)\right) = \nabla_\alpha \left(2\alpha^3 - 3\alpha^2 + \alpha\right) = \left(6\alpha^2 - 6\alpha + 1\right)$$

Solving for $6\alpha^2 - 6\alpha + 1 = 0$, we get the solutions:

$$\alpha = \left(\frac{3 + \sqrt{3}}{6}\right), \left(\frac{3 - \sqrt{3}}{6}\right)$$

Since both $(3 + \sqrt{3}/6), (3 - \sqrt{3}/6)$ lie between 0 and 1, we check for the second derivatives:

$$\nabla_\alpha^2 \left(\alpha(1 - \alpha)(1 - 2\alpha)\right) = \nabla_\alpha \left(6\alpha^2 - 6\alpha + 1\right) = 12\alpha - 6 = 6(2\alpha - 1)$$

At $\alpha = (3 + \sqrt{3})/6$, $\nabla_\alpha^2 = 6(2\alpha - 1) = 2\sqrt{3} > 0$.
At $\alpha = (3 - \sqrt{3})/6$, $\nabla_\alpha^2 = 6(2\alpha - 1) = -2\sqrt{3} < 0$.
Thus $\alpha = (3 + \sqrt{3})/6$ is a local minima, $\alpha = (3 - \sqrt{3})/6$ is a local maxima.
Substituting the two critical points into $\alpha(1 - \alpha)(1 - 2\alpha)$, we get $h_U = 9.623 \times 10^{-2}$, $h_L = -9.623 \times 10^{-2}$.
Thus, $h_U = 9.623 \times 10^{-2}$, $h_L = -9.623 \times 10^{-2}$ (for use in Theorem 3) and $g = 0.25$, $h = 0.09623$ (for use in Theorem 4).

### E.3 TANH ACTIVATION

For tanh activation, we have the following:

$$\sigma(x) = \tanh(x) = \frac{\exp(x) - \exp(-x)}{\exp(x) + \exp(-x)}$$
$$\sigma'(x) = (1 - \tanh(x))(1 + \tanh(x))$$
$$\sigma''(x) = -2\tanh(x)(1 - \tanh(x))(1 + \tanh(x))$$

The second derivative of tanh , i.e $(\sigma''(x))$ can be bounded using standard differentiation. Let $\alpha$ denote $\tanh(x)$. We know that $-1 \le \alpha \le 1$:

$$h_L \le \sigma''(x) \le h_U$$

$$h_L = \min_{0 \le \alpha \le 1} -2\alpha(1-\alpha)(1+\alpha)$$

$$h_U = \max_{0 \le \alpha \le 1} -2\alpha(1-\alpha)(1+\alpha)$$

To solve for both $h_L$ and $h_U$, we first differentiate $-2\alpha(1-\alpha)(1+\alpha)$ with respect to $\alpha$:

$$\nabla_\alpha \left(-2\alpha(1-\alpha)(1+\alpha)\right) = \nabla_\alpha \left(2\alpha^3 - 2\alpha\right) = \left(6\alpha^2 - 2\right)$$

Solving for $6\alpha^2 - 2 = 0$, we get the solutions:

$$\alpha = -\frac{1}{\sqrt{3}}, \frac{1}{\sqrt{3}}$$

Since both $-1/\sqrt{3}, 1/\sqrt{3}$ lie between -1 and 1, we check for the second derivatives:

$$\nabla_\alpha^2 \left(-2\alpha(1-\alpha)(1+\alpha)\right) = \nabla_\alpha \left(6\alpha^2 - 2\right) = 12\alpha$$

At $\alpha = -1/\sqrt{3}$, $\nabla_\alpha^2 = 12\alpha = -4\sqrt{3} < 0$.
At $\alpha = 1/\sqrt{3}$, $\nabla_\alpha^2 = 12\alpha = 4\sqrt{3} > 0$.
Thus $\alpha = 1/\sqrt{3}$ is a local minima, $\alpha = -1/\sqrt{3}$ is a local maxima.
Substituting the two critical points into $-2\alpha(1-\alpha)(1+\alpha)$, we get $h_U = 0.76981$, $h_L = -0.76981$.
Thus, $h_U = 0.76981$, $h_L = -0.76981$ (for use in Theorem 3) and $g = 1$, $h = 0.76981$ (for use in Theorem 4).

## F  QUADRATIC BOUNDS FOR TWO-LAYER RELU NETWORKS

For a 2 layer network with ReLU activation, such that the input $\mathbf{x}$ lies in the ball $\left\|\mathbf{x} - \mathbf{x}^{(0)}\right\| \le \rho$, we can compute the bounds over $\mathbf{z}^{(1)}$ directly:

$$\mathbf{W}_i^{(1)}\mathbf{x}^{(0)} + \mathbf{b}_i^{(1)} - \rho\left\|\mathbf{W}_i^{(1)}\right\| \le \mathbf{z}_i^{(1)} \le \mathbf{W}_i^{(1)}\mathbf{x}^{(0)} + \mathbf{b}_i^{(1)} + \rho\left\|\mathbf{W}_i^{(1)}\right\|$$

Thus we can get a lower bound and upper bound for each $\mathbf{z}_i^{(1)}$. We define $d_i$ and $u_i$ as the following:

$$d_i = \mathbf{W}_i^{(1)}\mathbf{x}^{(0)} + \mathbf{b}_i^{(1)} - \rho\left\|\mathbf{W}_i^{(1)}\right\| \tag{78}$$

$$u_i = \mathbf{W}_i^{(1)}\mathbf{x}^{(0)} + \mathbf{b}_i^{(1)} + \rho\left\|\mathbf{W}_i^{(1)}\right\| \tag{79}$$

We can derive the following quadratic lower and upper bounds for each $\mathbf{a}_i^{(1)}$:

$$\mathbf{a}_i^{(1)} \le \begin{cases} \dfrac{-d_i}{(u_i - d_i)^2}\left(\mathbf{z}_i^{(1)}\right)^2 + \dfrac{u_i^2 + d_i^2}{(u_i - d_i)^2}\mathbf{z}_i^{(1)} - \dfrac{u_i^2 d_i}{(u_i - d_i)^2} & |d_i| \le |u_i| \\[4mm] \dfrac{u_i}{(u_i - d_i)^2}\left(\mathbf{z}_i^{(1)}\right)^2 - \dfrac{2u_i d_i}{(u_i - d_i)^2}\mathbf{z}_i^{(1)} + \dfrac{u_i d_i^2}{(u_i - d_i)^2} & |d_i| \ge |u_i| \end{cases}$$

$$\mathbf{a}_i^{(1)} \ge \begin{cases} 0 & 2|d_i| \le |u_i| \\[2mm] \mathbf{z}_i^{(1)} & |d_i| \ge 2|u_i| \\[2mm] \dfrac{1}{u_i - d_i}\left(\mathbf{z}_i^{(1)}\right)^2 - \dfrac{d_i}{u_i - d_i}\mathbf{z}_i^{(1)} & \text{otherwise} \end{cases}$$

The above steps are exactly the same as the quadratic upper and lower bounds used in (Zhang et al., 2018a).
Using the above two inequalities and the identity:

$$\mathbf{z}_y^{(2)} - \mathbf{z}_t^{(2)} = \sum_{i=1}^{N_1} \left(\mathbf{W}_{y,i}^{(2)} - \mathbf{W}_{t,i}^{(2)}\right)\mathbf{a}_i^{(1)}$$

we can compute a quadratic lower bound for $\mathbf{z}_y^{(2)} - \mathbf{z}_t^{(2)}$ in terms of $\mathbf{z}_i^{(1)}$ by taking the lower bound for $\mathbf{a}_i^{(1)}$ when $\left(\mathbf{W}_{y,i}^{(2)} - \mathbf{W}_{t,i}^{(2)}\right) > 0$ and upper bound when $\left(\mathbf{W}_{y,i}^{(2)} - \mathbf{W}_{t,i}^{(2)}\right) <= 0$. Furthermore since $\mathbf{z}_i^{(1)} = \mathbf{W}_i^{(1)}\mathbf{x} + \mathbf{b}_i^{(1)}$, we can express the resulting quadratic in terms of $\mathbf{x}$. Thus, we get the following quadratic function :

$$\mathbf{z}_y^{(2)} - \mathbf{z}_t^{(2)} \geq \frac{1}{2}\mathbf{x}^T\mathbf{P}\mathbf{x} + \mathbf{q} + r$$

The coefficients $\mathbf{P}$, $\mathbf{q}$ and $r$ can be determined using the above procedure. Note that unlike in (Zhang et al., 2018a), RHS can be a non-convex function.

Thus, it becomes an optimization problem where the goal is to minimize the distance $1/2 \left\| \mathbf{x} - \mathbf{x}^{(0)} \right\|^2$ subject to RHS (which is quadratic in $\mathbf{x}$) being zero. That is both our objective and constraint are quadratic functions. In the optimization literature, this is called the S-procedure and is one of the few non-convex problems that can be solved efficiently (Boyd & Vandenberghe, 2004).

We start with two initial values called $\rho_{low}$ (initialized to 0) and $\rho_{high}$ (initialized to 5).

We start with an initial value of $\rho$, initialized at $1/2\left(\rho_{low} + \rho_{high}\right)$ to compute $d_i$ (eq. (78)) and $u_i$ (eq. (79)). If the final distance after solving the S-procedure is less than $\rho$, we set $\rho_{low} = \rho$. if the final distance is greater than $\rho$, we set $\rho_{high} = \rho$. Set new $\rho = 1/2\left(\rho_{low} + \rho_{high}\right)$. Repeat until convergence.

## G  ADDITIONAL EXPERIMENTS

Empirical accuracy means the fraction of test samples that were correctly classified after running a PGD attack (Madry et al., 2018) with an $l_2$ bound on the adversarial perturbations. Certified accuracy means the fraction of test samples that were classified correctly initially and had the robustness certificate greater than a pre-specified attack radius $\rho$. For both empirical and certified accuracy, we use $\rho = 0.5$. Unless otherwise specified, we use the class with the second largest logit as the attack target for the given input (i.e. the class $t$). All experiments were run on the MNIST dataset while noting that our results are scalable for more complex datasets. The notation $(L \times [1024], \text{activation})$ denotes a neural network with $L$ layers with the specified activation function, $(\gamma = c)$ denotes standard training with $\gamma$ set to $c$, (CRT, $c$) denotes CRT training with $\gamma = c$. Certificates CROWN and CRC are computed over 150 correctly classified images.

### G.1  RESULTS ON CONVOLUTIONAL NEURAL NETWORKS

We use a 2 layer (1 hidden layer network). We use 64 filters in the convolution layer and softplus activation function and stride of length 2. This is followed by reshaping and a fully connected layer.

| | MNIST | | | CIFAR-10 | | |
|---|---|---|---|---|---|---|
| $\gamma$ | Standard Accuracy | Empirical Robust Accuracy | Certified Robust Accuracy | Standard Accuracy | Empirical Robust Accuracy | Certified Robust Accuracy |
| 0 | 98.68% | 87.81% | 0.00% | 56.22% | 14.88% | 0.00% |
| 0.005 | 97.67% | 93.71% | **91.47%** | 56.18% | 30.19% | 9.37% |
| 0.01 | 97.08% | 92.92% | 91.25% | 53.52% | 31.82% | 17.39% |
| 0.02 | 96.36% | 90.98% | 89.58% | 49.55% | 31.80% | 25.93% |
| 0.03 | 95.54% | 89.99% | 88.75% | 46.56% | 31.98% | **29.26%** |

Table 8: Comparison between certified robust accuracy for different values of the regularization parameter $\gamma$ for a single hidden layer convolutional neural network with softplus activation function

## G.2 RESULTS FOR TANH NETWORKS

| Network | Training | Standard Accuracy | Empirical Robust Accuracy | Certified Robust Accuracy | Certificate (mean) | |
|---|---|---|---|---|---|---|
| | | | | | CROWN | CRC |
| 2×[1024], tanh | PGD | 98.76% | 95.79% | 84.11% | 0.30833 | 0.61340 |
| | TRADES | 98.63% | 96.20% | 93.72% | 0.40601 | 0.86287 |
| | CRT, 0.01 | 98.52% | 95.90% | **95.00%** | 0.37691 | **1.47016** |
| 3×[1024], tanh | PGD | 98.78% | 94.92% | 0.00% | 0.12706 | 0.03036 |
| | TRADES | 98.16% | 94.78% | 0.00% | 0.15875 | 0.02983 |
| | CRT, 0.01 | 98.15% | 95.00% | **94.16%** | 0.28004 | **1.14995** |
| 4×[1024], tanh | PGD | 98.53% | 94.53% | 0.00% | 0.07439 | 0.00140 |
| | TRADES | 97.08% | 92.85% | 0.00% | 0.11889 | 0.00068 |
| | CRT, 0.01 | 97.24% | 93.05% | **91.37%** | 0.33649 | **0.93890** |

Table 9: Comparison between CRT, PGD (Madry et al., 2018) and TRADES (Zhang et al., 2019) for Tanh networks. CRC outperforms CROWN significantly for 2 layer networks and when trained with our regularizer for deeper networks. CRT outperforms TRADES and PGD giving higher certified accuracy.

## G.3 COMPARING RANDOMIZED SMOOTHING AND TRADES WITH CRT

Randomized smoothing is designed to work in untargeted attack settings while CRT is for targeted attacks. Thus, to do a fair comparison of CRT with randomized smoothing, we make the following changes in randomized smoothing.

First, we use $n_0 = 100$ initial samples to select the label class ($l$) and false target class ($t$). The samples for estimation were $n = 100,000$ and failure probability was $\alpha = 0.001$. Then we use the binary version of randomized smoothing for estimation, i.e classify between $y$ and $t$. To find the adversarial example for adversarial training, we use the cross entropy loss for 2 classes ($y$ and $t$).

For TRADES, we select the class with second highest logit as the target class $t$ and use the 2 class version of the cross entropy loss for finding the adversarial example.

Table 10: Comparison between CRT and Randomized Smoothing(Cohen et al., 2019). $s$ denotes the standard deviation for smoothing. We use $\rho = 0.5$. For CRT, we use $\gamma = 0.01$

| Network | Randomized Smoothing | | | CRT |
|---|---|---|---|---|
| | $s = 0.25$ | $s = 0.50$ | $s = 1.0$ | _ |
| 2 × [1024], sigmoid | 93.75% | 93.09% | 88.91% | **95.61%** |
| 2 × [1024], tanh | 94.61% | 93.08% | 82.26% | **95.00%** |
| 3 × [1024], sigmoid | 94.00% | 93.03% | 86.58% | **94.99%** |
| 3 × [1024], tanh | 93.69% | 91.68% | 80.55% | **94.16%** |
| 4 × [1024], sigmoid | **93.68%** | 92.45% | 84.99% | 93.41% |
| 4 × [1024], tanh | **93.57%** | 92.19% | 83.90% | 91.37% |

Table 11: Comparison between CRC and CROWN-general (CROWN-Ada for relu) for different targets. For CRT training, we use $\gamma = 0.01$. We compare CRC with CROWN-general for different targets for 150 correctly classified images. Runner-up means class with second highest logit is considered as adversarial class. Random means any random class other than the label is considered adversarial. Least means class with smallest logit is adversarial. For 2-layer networks, CRC outperforms CROWN-general significantly even without adversarial training. For deeper networks (3 and 4 layers), CRC works better on networks that are trained with curvature regularization.

| Network | Training | Target | Certificate (mean) | | Time per Image (s) | |
|---|---|---|---|---|---|---|
| | | | CROWN | CRC | CROWN | CRC |
| $2 \times [1024]$, relu | standard | runner-up | 0.50110 | **0.59166** | 0.1359 | 2.3492 |
| | | random | 0.68506 | **0.83080** | 0.2213 | 3.5942 |
| | | least | 0.86386 | **1.04883** | 0.1904 | 3.0292 |
| $2 \times [1024]$, sigmoid | standard | runner-up | 0.28395 | **0.48500** | 0.1818 | 0.1911 |
| | | random | 0.38501 | **0.69087** | 0.1870 | 0.1912 |
| | | least | 0.47639 | **0.85526** | 0.1857 | 0.1920 |
| | CRT, 0.01 | runner-up | 0.43061 | **1.54673** | 0.1823 | 0.1910 |
| | | random | 0.52847 | **1.99918** | 0.1853 | 0.1911 |
| | | least | 0.62319 | **2.41047** | 0.1873 | 0.1911 |
| $2 \times [1024]$, tanh | standard | runner-up | 0.23928 | **0.40047** | 0.1672 | 0.1973 |
| | | random | 0.31281 | **0.52025** | 0.1680 | 0.1986 |
| | | least | 0.38964 | **0.63081** | 0.1726 | 0.1993 |
| | CRT, 0.01 | runner-up | 0.37691 | **1.47016** | 0.1633 | 0.1963 |
| | | random | 0.45896 | **1.87571** | 0.1657 | 0.1982 |
| | | least | 0.52800 | **2.21704** | 0.1697 | 0.1981 |
| $3 \times [1024]$, sigmoid | standard | runner-up | **0.24644** | 0.06874 | 1.6356 | 0.5012 |
| | | random | **0.29496** | 0.08275 | 1.5871 | 0.5090 |
| | | least | **0.33436** | 0.09771 | 1.6415 | 0.5056 |
| | CRT, 0.01 | runner-up | 0.39603 | **1.24100** | 1.5625 | 0.5013 |
| | | random | 0.46808 | **1.54622** | 1.6142 | 0.4974 |
| | | least | 0.51906 | **1.75916** | 1.6054 | 0.4967 |
| $3 \times [1024]$, tanh | standard | runner-up | **0.08174** | 0.01169 | 1.4818 | 0.4908 |
| | | random | **0.10012** | 0.01432 | 1.5906 | 0.4963 |
| | | least | **0.12132** | 0.01757 | 1.5888 | 0.5076 |
| | CRT, 0.01 | runner-up | 0.28004 | **1.14995** | 1.4832 | 0.4926 |
| | | random | 0.32942 | **1.41032** | 1.5637 | 0.4957 |
| | | least | 0.38023 | **1.65692** | 1.5626 | 0.4930 |
| $4 \times [1024]$, sigmoid | standard | runner-up | **0.19501** | 0.00454 | 4.7814 | 0.8107 |
| | | random | **0.21417** | 0.00542 | 4.6313 | 0.8377 |
| | | least | **0.22706** | 0.00609 | 4.7973 | 0.8313 |
| | CRT, 0.01 | runner-up | 0.40327 | **1.06208** | 4.1830 | 0.8088 |
| | | random | 0.47038 | **1.29095** | 4.3922 | 0.7333 |
| | | least | 0.52249 | **1.49521** | 4.4676 | 0.7879 |
| $4 \times [1024]$, tanh | standard | runner-up | **0.03554** | 0.00028 | 5.7016 | 0.8836 |
| | | random | **0.04247** | 0.00036 | 5.8379 | 0.8602 |
| | | least | **0.04895** | 0.00044 | 5.8298 | 0.9045 |
| | CRT, 0.01 | runner-up | 0.33649 | **0.93890** | 3.8815 | 0.8182 |
| | | random | 0.41617 | **1.18956** | 4.0013 | 0.8215 |
| | | least | 0.47778 | **1.41429** | 4.3856 | 0.8311 |

### G.4 MEASURING THE IMPACT OF CURVATURE REGULARIZATION

In Table 12, we measure how the standard accuracy, empirical accuracy, certified accuracy, upper bound on the curvature $K_{ub}$, lower bound on the curvature $K_{lb}$, changes as we increase the regularization parameter $\gamma$ and the network is trained with CRT.

In Table 13, we measure how the standard accuracy, empirical accuracy, certified accuracy, CROWN and CRC changes as we increase the regularization parameter $\gamma$ and the network is trained without any adversarial training.

### G.5 COMPUTING $K_{lb}$ AND $K_{ub}$

First, note that $K$ does not depend on the input, but on network weights $\mathbf{W}^{(I)}$, label $y$ and target $t$. Different images may still have different $K$ because label $y$ and target $t$ may be different.

To compute $K_{lb}$ in the table, first for each pair $y$ and $t$, we find the largest eigenvalue of the Hessian of all test images that have label $y$ and second largest logit of class $t$. Then we take the max of the largest eigenvalue across all test images. This gives a rough estimate of the largest curvature in the vicinity of test images with label $y$ and target $t$. We can directly take the mean across all such pairs to compute $K_{lb}$. However, we find that some pairs $y$ and $t$ were infrequent (with barely 1,2 test images in them). Thus, for all such pairs we cannot get a good estimate of the largest curvature in vicinity. We select all pairs $y$ and $t$ that have at least 100 images in them and compute $K_{lb}$ by taking the mean across all such pairs.

To compute $K_{ub}$ in the table, we compute $K$ for all pairs $y$ and $t$ that have at least 100 images, i.e at least 100 images should have label $y$ and target $t$. And then we compute the mean across all $K$ that satisfy this condition. This was done to do a fair comparison with $K_{lb}$.

Figure 2 shows a plot of the $K_{ub}$ and $K_{lb}$ with increasing $\gamma$ for a tanh network.

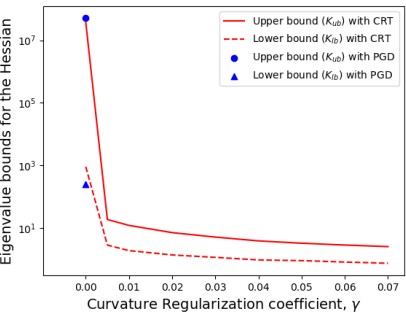

Figure 2: $K_{ub}$ and $K_{lb}$ are upper and lower curvature bounds of the network with Tanh activations (averaged over $(y, t)$ pairs). When $\gamma = 0$ (no curvature regularization), networks adversarially trained with CRT or PGD both have high curvatures. However, CRT with even a small $\gamma$ leads to a significant decrease in curvature bounds. Also, curvature bounds are higher with Tanh than with Sigmoid. Results are similar to Sigmoid in Figure 1.

Table 12: In this table, we measure the effect of increasing $\gamma$, when the network is trained with CRT on standard, empirical, certified robust accuracy, $K_{lb}$ and $K_{ub}$ (defined in subsection G.5) for different depths (2, 3, 4 layer) and activations (sigmoid, tanh). We find that for all networks $\gamma = 0.01$ works best. We find that the lower bound, $K_{lb}$ increases (for $\gamma = 0$) for deeper networks suggesting that deep networks have higher curvature. Furthermore, for a given $\gamma$ (say 0.005), we find that the gap between $K_{ub}$ and $K_{lb}$ increases as we increase the depth suggesting that $K$ is not a tight bound for deeper networks.

| Network | $\gamma$ | Standard Accuracy | Empirical Robust Accuracy | Certified Robust Accuracy | Curvature bound (mean) | |
|---|---|---|---|---|---|---|
| | | | | | $K_{lb}$ | $K_{ub}$ |
| 2×[1024], sigmoid | 0.0 | 98.77% | 96.17% | 95.04% | 7.2031 | 72.0835 |
| | 0.005 | 98.82% | 96.33% | **95.61%** | 3.8411 | 8.2656 |
| | 0.01 | 98.57% | 96.28% | **95.59%** | 2.8196 | 5.4873 |
| | 0.02 | 98.59% | 95.97% | 95.22% | 2.2114 | 3.7228 |
| | 0.03 | 98.30% | 95.73% | 94.94% | 1.8501 | 2.9219 |
| 2×[1024], tanh | 0.0 | 98.65% | 95.48% | 92.69% | 12.8434 | 107.5689 |
| | 0.005 | 98.71% | 95.88% | 94.76% | 4.8116 | 10.1860 |
| | 0.01 | 98.52% | 95.90% | **95.00%** | 3.4269 | 6.3529 |
| | 0.02 | 98.35% | 95.71% | 94.77% | 2.3943 | 4.1513 |
| | 0.03 | 98.29% | 95.39% | 94.54% | 1.9860 | 3.933 |
| 3×[1024], sigmoid | 0. | 98.52% | 90.26% | 0.00% | 19.2131 | 3294.9070 |
| | 0.005 | 98.41% | 95.81% | 94.91% | 2.6249 | 13.4985 |
| | 0.01 | 98.23% | 95.70% | **94.99%** | 1.9902 | 8.6654 |
| | 0.02 | 97.99% | 95.33% | 94.64% | 1.4903 | 5.4380 |
| | 0.03 | 97.86% | 94.98% | 94.15% | 1.2396 | 4.1409 |
| | 0.04 | 97.73% | 94.60% | 93.88% | 1.0886 | 3.3354 |
| | 0.05 | 97.60% | 94.45% | 93.65% | 0.9677 | 2.7839 |
| 3×[1024], tanh | 0. | 98.19% | 86.38% | 0.00% | 133.7992 | 17767.5918 |
| | 0.005 | 98.13% | 94.56% | 93.01% | 3.2461 | 17.5500 |
| | 0.01 | 98.15% | 95.00% | **94.16%** | 2.2347 | 10.8635 |
| | 0.02 | 97.84% | 94.79% | 94.05% | 1.6556 | 6.7072 |
| | 0.03 | 97.70% | 94.19% | 93.42% | 1.3546 | 5.0533 |
| | 0.04 | 97.57% | 94.04% | 92.95% | 1.1621 | 4.0071 |
| | 0.05 | 97.31% | 93.66% | 92.65% | 1.0354 | 3.3439 |
| 4×[1024], sigmoid | 0. | 98.22% | 83.04% | 0.00% | 86.9974 | 343582.3125 |
| | 0.005 | 98.18% | 95.02% | 93.20% | 2.1760 | 15.3358 |
| | 0.01 | 97.83% | 94.65% | **93.41%** | 1.6823 | 10.2289 |
| | 0.02 | 97.33% | 94.02% | 92.94% | 1.2089 | 6.5573 |
| | 0.03 | 97.07% | 93.52% | 92.65% | 1.0144 | 4.9576 |
| | 0.04 | 96.70% | 92.78% | 91.95% | 0.8840 | 3.9967 |
| | 0.05 | 96.38% | 92.29% | 91.33% | 0.7890 | 3.4183 |
| | 0.06 | 96.29% | 92.17% | 91.11% | 0.7128 | 3.0050 |
| | 0.07 | 96.08% | 91.83% | 90.67% | 0.6614 | 2.6905 |
| 4×[1024], tanh | 0. | 97.45% | 75.18% | 0.00% | 913.6984 | 37148156 |
| | 0.005 | 97.48% | 93.29% | 89.98% | 2.8690 | 18.8079 |
| | 0.01 | 97.24% | 93.05% | **91.37%** | 1.9114 | 12.2148 |
| | 0.02 | 96.82% | 92.65% | 91.35% | 1.3882 | 7.1771 |
| | 0.03 | 96.27% | 91.43% | 90.09% | 1.1643 | 5.1671 |
| | 0.04 | 95.62% | 90.69% | 89.41% | 0.9620 | 3.9061 |
| | 0.05 | 95.77% | 90.69% | 89.40% | 0.9160 | 3.2909 |
| | 0.06 | 95.52% | 90.00% | 88.38% | 0.8234 | 2.8808 |
| | 0.07 | 95.24% | 89.51% | 87.91% | 0.7540 | 2.5635 |

Table 13: In this table, we measure the impact of increasing curvature regularization ($\gamma$) on accuracy, empirical robust accuracy, certified robust accuracy, CROWN-general and CRC when the network is trained without any adversarial training. We find that adding a very small amount of curvature regularization has a minimal impact on the accuracy but significantly increases CRC. Increase in CROWN certificate is not of similar magnitude. Somewhat surprisingly, we observe that even without any adversarial training, we can get nontrivial certified accuracies of $84.73\%, 88.66\%, 89.61\%$ on 2,3,4 layer sigmoid networks respectively.

| Network | $\gamma$ | Standard Accuracy | Empirical Robust Accuracy | Certified Robust Accuracy | Certificate (mean) | |
| --- | --- | --- | --- | --- | --- | --- |
| | | | | | CROWN | CRC |
| | 0. | 98.37% | 76.28% | 54.17% | 0.28395 | 0.48500 |
| | 0.005 | 97.96% | 88.65% | 82.68% | 0.36125 | 0.83367 |
| $2 \times [1024]$, sigmoid | 0.01 | 98.08% | 88.82% | 83.53% | 0.32548 | 0.84719 |
| | 0.02 | 97.88% | 88.90% | 83.68% | 0.34744 | 0.86632 |
| | 0.03 | 97.73% | 89.28% | **84.73%** | 0.35387 | 0.90490 |
| | 0. | 98.34% | 79.10% | 14.42% | 0.23938 | 0.40047 |
| | 0.005 | 98.01% | 89.95% | 85.70% | 0.27262 | 0.89672 |
| $2 \times [1024]$, tanh | 0.01 | 97.99% | 90.17% | 86.18% | 0.28647 | 0.93819 |
| | 0.02 | 97.64% | 90.13% | **86.40%** | 0.30075 | 0.99166 |
| | 0.03 | 97.52% | 89.96% | 86.22% | 0.30614 | 0.98771 |
| | 0. | 98.37% | 85.19% | 0.00% | 0.24644 | 0.06874 |
| | 0.005 | 97.98% | 91.93% | **88.66%** | 0.38030 | 0.99044 |
| | 0.01 | 97.71% | 91.49% | 88.33% | 0.39799 | 1.07842 |
| $3 \times [1024]$, sigmoid | 0.02 | 97.50% | 91.34% | 88.38% | 0.38091 | 1.08396 |
| | 0.03 | 97.16% | 91.10% | 88.63% | 0.41015 | 1.15505 |
| | 0.04 | 97.03% | 90.96% | 88.48% | 0.42704 | 1.18073 |
| | 0.05 | 96.76% | 90.65% | 88.30% | 0.43884 | 1.19296 |
| | 0. | 97.91% | 77.40% | 0.00% | 0.08174 | 0.01169 |
| | 0.005 | 97.45% | 91.32% | **88.57%** | 0.28196 | 0.95367 |
| | 0.01 | 97.29% | 90.98% | 88.31% | 0.31237 | 1.05915 |
| $3 \times [1024]$, tanh | 0.02 | 97.04% | 90.21% | 87.77% | 0.30901 | 1.08607 |
| | 0.03 | 96.88% | 90.02% | 87.52% | 0.34148 | 1.11717 |
| | 0.04 | 96.53% | 89.61% | 86.87% | 0.36583 | 1.11307 |
| | 0.05 | 96.31% | 89.25% | 86.26% | 0.38519 | 1.11689 |
| | 0. | 98.39% | 83.27% | 0.00% | 0.19501 | 0.00454 |
| | 0.005 | 97.74% | 91.67% | 88.95% | 0.36863 | 0.91840 |
| | 0.01 | 97.41% | 91.71% | **89.61%** | 0.40620 | 1.05323 |
| | 0.02 | 96.47% | 90.03% | 87.77% | 0.45074 | 1.14219 |
| $4 \times [1024]$, sigmoid | 0.03 | 96.24% | 90.40% | 88.14% | 0.47961 | 1.30671 |
| | 0.04 | 95.65% | 89.61% | 87.54% | 0.49987 | 1.35129 |
| | 0.05 | 95.36% | 89.10% | 87.09% | 0.51187 | 1.36064 |
| | 0.06 | 95.29% | 88.96% | 87.01% | 0.52629 | 1.38666 |
| | 0.07 | 95.23% | 88.03% | 85.93% | 0.54754 | 1.27948 |
| | 0. | 97.65% | 69.20% | 0.00% | 0.03554 | 0.00028 |
| | 0.005 | 97.02% | 89.77% | 85.98% | 0.29410 | 0.82364 |
| | 0.01 | 96.52% | 89.38% | **86.40%** | 0.34778 | 0.97365 |
| | 0.02 | 96.09% | 88.79% | 86.09% | 0.41662 | 1.10860 |
| $4 \times [1024]$, tanh | 0.03 | 95.74% | 88.36% | 85.65% | 0.44981 | 1.17400 |
| | 0.04 | 95.10% | 87.50% | 84.74% | 0.48356 | 1.21957 |
| | 0.05 | 95.14% | 87.72% | 84.77% | 0.49113 | 1.25076 |
| | 0.06 | 94.66% | 86.96% | 84.28% | 0.51104 | 1.28653 |
| | 0.07 | 94.34% | 86.67% | 83.90% | 0.49750 | 1.24198 |

### G.6 COMPARING OUR ATTACK AGAINST $l_2$ BOUNDED PGD

In this section, we empirically compare our Curvature-based attack optimization to the PGD method of Madry et al. (2018) (200 steps of size $0.01$). We should note that the objectives of our attack and PGD are different. The PGD attack (with sufficiently large number of steps) will always find a solution $\mathbf{x}^{(attack)}$ such that $\|\mathbf{x}^{(attack)} - \mathbf{x}^{(0)}\|_2 \approx \rho$. However, if our curvature bound is loose (or even if the curvature bound is tight but the network itself has large curvature), our method may converge to a point inside the ball where $\|\mathbf{x}^{(attack)} - \mathbf{x}^{(0)}\|_2 < \rho$. The primary benefit of our method is that if it converges to a point on the surface of the ball (i.e. $\|\mathbf{x}^{(attack)} - \mathbf{x}^{(0)}\|_2 = \rho$), then $\mathbf{x}^{(attack)}$ is provably the worst case perturbation in the $l_2$ ball of radius $\rho$ around the input $\mathbf{x}^{(0)}$ (since primal=dual using Theorem 2).

Thus, to have the same perturbation magnitude between our attack optimization and PGD, we consider a subset of samples where our attack method converges to a point on the surface of the ball. In Table 14, we show a comparison between average values of $\mathbf{z}_y - \mathbf{z}_t$ computed using PGD and our method where $y$ is the correct label and $t$ is the attack target. In this table, Mean CBA (Curvature-Based Attack) is computed by taking the average of $\mathbf{z}_y - \mathbf{z}_t$ for all test samples satisfying $\|\mathbf{x}^{(attack)} - \mathbf{x}^{(0)}\|_2 = \rho$ (with curvature based attack optimization). The relative gain is computed as (Mean PGD - Mean CBA)/(Mean PGD).

Table 14: Comparison between mean values of $\mathbf{z}_y - \mathbf{z}_t$ for $l_2$ bounded PGD and Curvature-Based Attack (CBA) optimization. See the experimental details in Section G.6.

| Network | Mean PGD | Mean CBA | Relative Gain |
|---|---|---|---|
| $2 \times [1024]$, sigmoid | 5.1526 | 4.9296 | 4.33% |
| $3 \times [1024]$, sigmoid | 3.7748 | 3.6342 | 3.72% |
| $4 \times [1024]$, sigmoid | 3.2024 | 3.0846 | 3.68% |

Table 15: Table showing attack success rates for different values of $\gamma$. Attack success rate denotes the fraction of points ($\mathbf{x}^{(0)}$) satisfying $\|\mathbf{x}^{(attack)} - \mathbf{x}^{(0)}\|_2 = \rho$ implying primal = dual in Theorem 2.

| Network | $\gamma$ | Attack success rate |
|---|---|---|
| $2 \times [1024]$, sigmoid | 0. | 5.05% |
| | 0.01 | 100% |
| | 0.02 | 100% |
| $3 \times [1024]$, sigmoid | 0. | 0.% |
| | 0.01 | 44.86% |
| | 0.03 | 100% |
| | 0.05 | 100% |
| $4 \times [1024]$, sigmoid | 0. | 0.% |
| | 0.01 | 24.42% |
| | 0.03 | 88.82% |
| | 0.05 | 99.97% |

