# OpenReview forum: "Curvature-based Robustness Certificates against Adversarial Examples"
_ICLR.cc/2020/Conference — Reject_

### Official Review · AnonReviewer3 · 2019-10-18
**Official Blind Review #3**

**Rating:** 6

**Review:**

Summary
========
This paper proposes the Curvature-based Robustness Certificate (CRC) and Curvature-based Robust Training (CRT) for robustness certificate against adversarial examples. The proposed techniques are theoretically formulated and empirically justified. The authors showed that, when the curvature (Hessian) of the network is bounded, improved certificate can be achieved by convex optimization. Explicit curvature regularization via CRT seems further improve both the certified robustness accuracy and the certificate, at a cost of 2% decrease in empirical robust accuracy.

While the proposed approaches are theoretically sound, I have several concerns, mostly on the experiments.
=================
1. Existing certification methods should be compared in more details, in terms of different assumptions, activation functions, etc. A summary table can help.

2. Comprehensive comparisons to more existing works such as "Certified adversarial robustness via randomized smoothing" (more formal comparison than in the appendix).

3. Experiments on CIFAR-10 and different activation functions in the main text.

4. Not sure why compare to uncertified defenses PGD and TRADES, though it doesn't seem to hurt the conclusions.

===============
After rebuttal:

Thanks for the response, my concerns have been addressed, rating upgraded to 6: weak accept.

**Experience Assessment:**

I have published one or two papers in this area.

**Review Assessment: Checking Correctness Of Derivations And Theory:**

I assessed the sensibility of the derivations and theory.

**Review Assessment: Checking Correctness Of Experiments:**

I carefully checked the experiments.

**Review Assessment: Thoroughness In Paper Reading:**

I read the paper at least twice and used my best judgement in assessing the paper.

---

> ### Author Response · Authors · 2019-11-08
> **Updated the paper**
>
> Thank you for your comments. Below find our responses:
>
> 1) We now have added a table to the main text (Table 2) that compares the setup used for different robustness certifications including ours.
>
> 2) We have included comparison with the smoothing-based robustness certificate in Appendix Table 10. Our results show that our method gives higher robust accuracy than Randomized Smoothing for 2 and 3 layer networks (trained with curvature regularization). We note that there is a fundamental difference between our certificate (and CROWN) with the smoothing one since our certificate is deterministic while the smoothing-based certificate is probabilistic (holds with high probability). In the revised version, we explain this difference more explicitly in the main text.
>
> 3) Now we have updated the paper to include some experimental evaluation on the CIFAR-10 dataset in Table 6 (in the main text) and Table 8 (in the appendix). We observe a similar trend to our previous MNIST experimental results where our certificate on regularized networks significantly outperforms the existing method CROWN. We note that in experiments of Table 6, we use a fully connected shallow neural network which is known to not give high accuracy on CIFAR-10. However, our certification results can be used for other network architectures such as convolutional networks. To demonstrate this, in experiments of Table 8, we use a two layer convolutional neural network which improves the accuracy on CIFAR-10 dataset.
>
>
> 4) We included comparison with TRADES and PGD because it is assumed that they provide robust models without actually proving that the model is robust. In summary, for a 2 layer network we observe that PGD and TRADES provide certified accuracy only slightly below our method (with curvature regularization) but for deeper networks our method gives significantly higher certified robust accuracy. We thought this observation can be of interest to the readers.

---

> ### Author Response · Authors · 2019-11-12
> **Follow up**
>
> Thank you again for your time and constructive comments. Please let us know if our responses have addressed your comments or if you have any additional questions. Thank you!

---

### Official Review · AnonReviewer2 · 2019-10-25
**Official Blind Review #2**

**Rating:** 6

**Review:**

This paper gives a global curvature bound for general neural networks, and used this bound for certifying robustness of neural networks. The basic techniques include writing the neural network verification problem as a constrained optimization problem (similar to [1]), and construct a Lagrangian dual for it, and discuss the situation where the dual problem is convex and can be solved efficiently. The global curvature bound is the crucial step to determine the condition that the optimization problem is convex.

The benefits of the proposed methods include efficiency (bounds m and M only need to be solved for each target label pair, and the optimization process to get the certificate is fast), and tight certified accuracy for shallow networks. Also, it can be used during training to improve the tightness of CRC bounds.  There are also a few weakness: the certificate unfortunately only works for non-ReLU networks, and only for L2 norm; for networks beyond 2 layers, CRC cannot outperform CROWN unless it is specially trained using the CRC loss.

Questions and concerns:

1. For the curvature regularization training, some comparisons against other certified defenses are needed - PGD and TRADES are not good examples since they are not certified defense methods, and 0% certified accuracy are obtained. I understand many existing works like Wong et al., 2018 only work on ReLU networks, but I believe interval bound propagation (IBP) based methods [2] should be easily applicable to any monotonic activation functions. At least, the authors should compare verified accuracy at the same epsilon settings (\pho=1.58) as in Wong et al., 2018.  In page 22 (last page of appendix in arxiv version) of [3], you can find their L2 robustness training results.  How does the certified accuracy of CRC compare to these results?

2. The paper mentions the Attack problem and proposes an attack algorithm (Algorithm 2), however I am not able to find any experiments on the attack. If the authors want to claim the contribution of curvature based attacks, some empirical results should be given, and at least compare it to a 200-step PGD baseline and see which one finds smaller adversarial examples.

3. The claim (in introduction and conclusions) that CRC is much faster than CROWN probably won't hold under a fair comparison. I believe the gradient decent based certification algorithm (Algorithm 1) was computed efficiently on GPU, where the NN is defined. For a fair comparison, it should be also compared to a GPU implementation of CROWN ([4] provides such an implementation).  In my experience, CROWN is an efficient algorithm that can be even used iteratively during training (as a certified defense [4]), so for the small networks used in this paper, it should compute bounds almost instantly on GPUs. I think it is better to revoke this claim, and in experiments the authors should clearly state that CROWN was computed on CPU so time is not comparable.

4. Also the claim "CRC outperforms CROWN's certificate significantly" should be made clearer that it only holds for CRC trained models. According to Table 3, CRC is significantly worse than CROWN if the model is not CRC trained.  Additionally, it should be made clear that the proposed method currently only applies to L2 norm setting.

5. (Minor) There are several duplicated references, e.g., on page 9, Cohen et al., (randomized smoothing) was cited twice as two different papers; the same is with Madry et al., on page 10, and Zhang et al. (CROWN) on page 11. These causes some confusions, e.g., on page 2, when talking about the bounded Lipschitz constant, I believe the correct citation for (Zhang et al., 2018b) should be another paper [5] from the same first author which is on bounding Lipschitz constant.

Improvements and extensions:

I think the proposed method can be greatly improved by using "local curvature", where the curvature bounds m and M are computed within a local region near an input point. This is sufficient as long as our optimization does not escape this safe region, and the safe region can be naturally defined as the perturbation radius to be certified. The local curvature can be obtained by giving a tighter bounds on the second derivative of activation function, rather than considering just the worst case. Using CROWN, we can obtain pre-activation upper and lower bounds. These bounds can be used to bound the second derivative. For example, if a tanh neuron's input is bounded by -0.1 and 0.1 (those bounds can be obtained efficiently by CROWN), the second derivative of tanh is bounded between -0.197356 and +0.197356, much better than the worst case bound -0.76981 and +0.76981 used in current global curvature bound. A similar technique to bound Jacobian matrix was used in [5]. Including some results for local curvature certificates will greatly increase the contribution of this paper, and make it a complete work. I strongly encourage the authors to do so, and feel free to discuss with me on any questions.


Despite some concerns, the main contribution of giving global curvature bounds of neural networks is valid. Thus I vote for accepting this paper, however the authors should make sure to address all the concerns.

[1] Salman, Hadi, et al. "A convex relaxation barrier to tight robust verification of neural networks." arXiv preprint arXiv:1902.08722 (2019).
[2] Gowal, Sven, et al. "On the effectiveness of interval bound propagation for training verifiably robust models." arXiv preprint arXiv:1810.12715 (2018).
[3] Wong, Eric, et al. "Scaling provable adversarial defenses." Advances in Neural Information Processing Systems. 2018. https://arxiv.org/pdf/1805.12514.pdf
[4] Zhang, Huan, et al. "Towards Stable and Efficient Training of Verifiably Robust Neural Networks." arXiv preprint arXiv:1906.06316 (2019).
[5] Zhang, H., Zhang, P., & Hsieh, C. J. Recurjac: An efficient recursive algorithm for bounding jacobian matrix of neural networks and its applications. arXiv preprint arXiv:1810.11783 (2018).


**Experience Assessment:**

I have published in this field for several years.

**Review Assessment: Checking Correctness Of Derivations And Theory:**

I assessed the sensibility of the derivations and theory.

**Review Assessment: Checking Correctness Of Experiments:**

I carefully checked the experiments.

**Review Assessment: Thoroughness In Paper Reading:**

I read the paper thoroughly.

---

> ### Author Response · Authors · 2019-11-08
> **Updated the paper to address your comments**
>
> Thank you for your detailed and extremely helpful feedback. We have updated the paper to address your comments. Below we summarize our responses:
>
> 1) Now we have included a table (Table 5 in the main text) comparing our results with those of Wong et al. 2018 (ref [3] in your comment) at the same epsilon setting (\rho=1.58). First, we note that the activation functions used in this comparison are different because the method proposed in Wong et al 2018 uses ReLU while our proposed CRT can only work with fully differentiable activation functions. Thus in our comparison, we use softplus activation function since it is a smoothed version of ReLU. Over a network trained with curvature regularization, we observe significant improvements over the other method for 2 and 3 layer networks while for a 4 layer network, we achieve a slightly lower accuracy (0.22% lower).
>
> We note that although the IBP bounds can potentially be extended to other monotonic activations, we could not find any existing github implementations with differentiable activation functions such as softplus to directly compare with our results.
>
> 2) First note that our attack method is fundamentally different from the existing attack methods such as l2 bounded PGD in that the other attack methods are guaranteed to converge at the boundary of the l2 ball. However, in the case of our curvature-based attack optimization, the convergence depends on the curvature bound m (Definition 2) because the lagrange multiplier eta is constrained to be larger than -m. Thus if -m is very large, the objective function heavily penalizes the (\|x-x^(0)\|^{2}-\rho) term and the optimization will converge to a point close to the input. However, the benefit of our curvature-based attack optimization is that unlike existing attack methods, if our method does converge to a point at the boundary of the l2 ball, then we are provably at the worst case perturbation inside the given l2 ball (by Theorem 2, primal=dual in this case). There is no such guarantee in the existing attack methods.
>
> Given the above explanation, now we have included a comparison with the l2 bounded PGD (200 steps) in the Appendix Table 14. We explain the experimental setup in Appendix Section G.6. In order to do a fair comparison, we only consider cases where the perturbation magnitude of converged points of our attack and PGD are the same (\rho=0.5). We observe that our method provides roughly 4.3%, 3.7% and 3.6% gain over PGD in 2,3, and 4 layer networks, respectively. Gain is measured as the relative decrease of the loss function.
>
> 3) Both CRC and CROWN were implemented on the CPU to compute their running times. However, we use a batched implementation of CRC i.e our CPU implementation of CRC computes the certificate for a batch of images while the CROWN implementation computes the certificate for one image at a time. Thus, we notice that these running time numbers are not directly comparable. We now explicitly mention this in the paper (in the caption of Table 3 in the main text) and avoid making conclusions regarding the running times because of the aforementioned differences in the implementations.
>
> 4) Now we explicitly mention in the abstract and the main text that CRC outperforms CROWN only when trained with a curvature regularizer. We also make it explicit in the abstract and the main text (Table 2) that our proposed certificate works with the l2 norm.
>
> 5) We have removed the duplicated references and added the correct citation for "bounded lipschitz constant".
>
>
> Additional improvements and extensions:
>
> Thank you very much for the suggestion regarding the local vs global curvature bounds. In fact, as you mentioned, using local curvature bounds can improve our certificate significantly. In the revised draft, we have added a section 7.1 in the main text to explain this. For a two-layer network, we have added experimental results in Table 7 that shows significant improvements in CRC using the local curvature bound. For deeper networks, however, computing local curvature bounds is more challenging and we leave exploring this direction for the future work.

---

> > ### Comment · AnonReviewer2 · 2019-11-14
> > **Thank you for the response. Some follow-up questions.**
> >
> > Thank you for the response.  The new experiments on models trained using Wong et al. looks promising, and I am glad to see that local curvature bounds improve the results.
> >
> > Regarding the comparison to IBP, the IBP code base provided by DeepMind[1] does support many activation functions including softplus. In fact any monotonic activation functions can be implemented very easily[2]. Also, [3] provides IBP training code for L2 norm. A comparison to IBP will greatly increase the contribution of this paper, as IBP achieves SOTA in many settings.
> >
> > For the attack part, thank you for the explanations and providing additional experiments in Appendix. Can you also report attack success rates? The table reports a relative improvement in reducing loss function and it seems not very intuitive. You can train a robust network using adversarial training and apply your attack, and see how many more examples you can attack compared to PGD. Also it is good to report the percentage of examples where you can guarantee that the found adversarial example is the worst case adversarial example, which is a main benefit of your attack.
> >
> > For the local curvature bounds, why it is more challenging for multiple layers? I think you can just apply your existing algorithm on multiple layers with a tighter second derivative bound obtained by CROWN?
> >
> > My main concern of this paper is that the proposed certificate seems tight only when it is trained - in that sense it is similar to IBP, where the bound can be very loose if not trained, but becomes significantly tighter after training. The proposed method seems to be much less effective and less efficient than IBP (higher computational cost, not so successful for networks with more that 3 layers, and not applicable to commonly used ReLU activation functions). On the other hand, if you train the CROWN bound, you can get much better certificates as well. So the benefits and use case of the proposed method can be quite limited, or at least, not very well demonstrated since there is no comparison to trained IBP/CROWN bounds.
> >
> > Overall, I tend to accept this paper despite its limitations. It is a good attempt of using curvature information to provide robustness certifications, although the results are not very strong and its usefulness is limited.

---

> > > ### Author Response · Authors · 2019-11-14
> > > **Response to follow up questions**
> > >
> > > Thank you for your response.
> > >
> > > The code provided by Deepmind is in Tensorflow while the code by Wong et al. is in Pytorch. We contacted the lead author of [3] and he confirmed that he is not aware of any existing implementation for IBP training with continuous activations. Given the short period of the rebuttal, it was not possible for us to implement and merge these two and implement a baseline for measuring robustness against l2 attacks. However, our results in Table 5 highlights a significant improvement of our certificate against a trained-network for IBP for 2 and 3 layer networks (although with different activations). In the camera ready version, we will include a comparison with IBP training with the same activation functions.
> > >
> > > We have updated the paper and included Table 15 reporting the attack success rate; the fraction of samples that our attack method provably finds the worst case perturbations for them. Interestingly, we observe that for a 2 layer network, a very small amount of curvature regularization (0.01) leads to 100% attack success rate. For deeper networks, we need higher amount of regularization to achieve similar success rate (0.03 for a 3 layer network and 0.05 for a 4 layer network).
> > >
> > > For deeper networks, indeed we can use CROWN to get local bounds on the first and second derivatives of individual activation functions (g and h in Theorem 4). However, in order to use the current approach that we are proposing to bound the curvatures, one needs to consider the maximum of the upper bounds and the minimum of the lower bounds across all activations in all layers.  We tried this trivial approach and it did not give significant gains compared to the current global bounds we have. A better approach is to use the individual bounds on the first- and second order derivatives in order to compute the curvature bounds. A direct generalization of theorem 4 for this case is possible wherein we simply bound the first and second derivative of each individual activation in the network to derive a local curvature bound. But given the short rebuttal period, we did not have enough time to engineer such an approach and leave this for future work.
> > >
> > > We agree that our robustness certification is tight only when we are adding curvature regularization during the training (for > 2 layer networks). However, as you mentioned, this is the case for all existing certification approaches such as IBP and smoothing. Moreover, during the training, we do not directly maximize the proposed certificate. Instead we only employ a more principled version of the commonly used curvature regularization. Curvature regularization is a standard technique that has been shown to achieve high empirical robustness on large sized neural networks [1, 2]. Our paper provides theoretical underpinnings for the same idea. Moreover, existing papers [1, 2] use approximations to reduce the curvature. For example, [1] uses the direction of the gradient as the high curvature direction and use a finite difference based approximation to reduce the curvature. We, however, derive a closed form expression for the Hessian and a global bound on the curvature of the network and use that to regularize the network to have small curvature. This also suggests that existing methods of curvature regularization (for empirical robustness) can be further improved using the ideas from this paper to boost their certified robustness.
> > >
> > > At the end, we thank you again for your time and very helpful comments which we believe improved our paper significantly.
> > >
> > > [1] https://arxiv.org/abs/1811.09716
> > > [2] https://arxiv.org/abs/1907.02610
> > > [3] https://arxiv.org/abs/1805.12514

---

> ### Author Response · Authors · 2019-11-12
> **Follow up**
>
> Thank you again for your time and constructive comments. Please let us know if our responses have addressed your comments or if you have any additional questions. Thank you!

---

### Official Review · AnonReviewer1 · 2019-10-28
**Official Blind Review #1**

**Rating:** 6

**Review:**

This paper develops computationally-efficient convex relaxations for robustness certification and adversarial attack problems given the classifier has a bounded curvature. The authors showed that the convex relaxation is tight under some general conditions. To be able to use proposed certification and attack convex optimizations, the authors derive global curvature bounds for deep networks with differentiable activation functions. The result is a consequence of a closed-form expression that the paper derived for the Hessian of a deep network.

The empirical results indicate that the proposed curvature-based robustness certificate outperforms the CROWN certificate by an order of magnitude while being faster to compute as well. Furthermore, adversarial training using the attack method coupled with curvature regularization results in a significantly higher certified robust accuracy than the existing adversarial training methods.

1.  I am not at all familiar with the robustness certification, therefore little knowledge of relevant related works.

2. I hope the authors can provide a more thorough survey of related works.

3. There exist good amount of theoretical analysis in the paper. However all empirical results are only from MNIST. This certainly weakens the method's contribution.


**Experience Assessment:**

I have read many papers in this area.

**Review Assessment: Checking Correctness Of Derivations And Theory:**

I assessed the sensibility of the derivations and theory.

**Review Assessment: Checking Correctness Of Experiments:**

I assessed the sensibility of the experiments.

**Review Assessment: Thoroughness In Paper Reading:**

I read the paper at least twice and used my best judgement in assessing the paper.

---

> ### Author Response · Authors · 2019-11-08
> **Updated the paper**
>
> Thank you for your helpful comments. We have updated the paper to address your comments.
>
> We have included a thorough survey of related works in Appendix A and cited all the relevant papers in the introduction section as well.
>
> Now we have updated the paper to include some experimental evaluation on the CIFAR-10 dataset in Table 6 (in the main text) and Table 8 (in the appendix). We observe a similar trend to our previous MNIST experimental results where our certificate on regularized networks significantly outperforms the existing method CROWN. We note that in experiments of Table 6, we use a fully connected shallow neural network which is known to not give high accuracy on CIFAR-10. However, our certification results can be used for other network architectures such as convolutional networks. To demonstrate this, in experiments of Table 8, we use a two layer convolutional neural network which improves the accuracy on CIFAR-10 dataset.

---

> > ### Author Response · Authors · 2019-11-12
> > **Follow up**
> >
> > Thank you again for your time and constructive comments. Please let us know if our responses have addressed your comments or if you have any additional questions. Thank you!

---

### Decision · Program_Chairs · 2019-12-19

**Decision:**

Reject

**Comment:**

This paper presents a upper bound on the curvature of a deep network. After the discussion, the author has addressed some concerns of reviwers, but the results are not very strong, there is some limitation on the applications. There is no strong support for this paper. Due to the high standard of ICLR, the acceptance of the paper need strong results in terms of theory or experiments.